# Liquid–liquid phase separation of LARP7 restrains HIV-1 replication

Zhuoxin Li [1], Xiya Fang[1], Bing Zhao[1], Ran Liu[1], Yezhuang Shen[1], Tingting Li[1], Yining Wang[1], Zenglin Guo[1], Wen Wang[1], Biyu Zhang[1], Qiuying Han[1], Xin Xu[1], Kai Wang[1], Libing Yin[1], Weili Gong[1], Ailing Li[1,2,3], Tao Zhou[1,2], Teng Li [1✉] & Weihua Li [1✉]

## Abstract

HIV-1 initiates replication by its transactivator Tat, hijacking the positive transcription elongation factor b (P-TEFb) in the host cell. Most P-TEFb is maintained in an inactive state by 7SK snRNP until it is brought to the transcription initiation complex by cellular or viral transactivators that accelerate transcription and facilitate the production of full-length viral transcripts. Here, we report that HIV-1 infection triggers liquid-liquid phase separation of LARP7, a central component of 7SK snRNP. Tat is incorporated into HIV-1-induced LARP7 condensates after infection. Conserved lysine residues in the intrinsically disordered region of LARP7 are essential for both its phase separation and the inhibition of Tat-mediated transcription. These findings identify a mechanism wherein P-TEFb and Tat are sequestered within LARP7 condensates, restraining HIV-1 transcription.

Key words HIV-1; La-Related Protein 7; Liquid–Liquid Phase Separation; Tat
Subject Categories Chromatin, Transcription & Genomics; Microbiology, Virology & Host Pathogen Interaction; Signal Transduction

## Introduction

Human immunodeficiency virus type 1 (HIV-1) poses a severe threat to the immune system, ultimately progressing to acquired immune deficiency syndrome (AIDS), a critical global health challenge for which no definitive cure currently exists (Landovitz et al, 2023). The 9.8-kb genome of HIV-1 encodes only three structural genes (*gag*, *pol*, *env*), two regulatory genes (*Tat*, *Rev*), and four helper genes (*nef*, *vpr*, *vpu*, *vif*). Consequently, HIV-1 relies heavily on host cellular proteins to complete its life cycle, rendering these proteins critical targets for therapeutic intervention (Dürr et al, 2015). The viral transactivator Tat plays a central role in regulating the replication of HIV-1 (Ott et al, 2011). The human positive transcription elongation factor b (P-TEFb), a heterodimer

of CDK9 and Cyclin T1/2, is an essential host cofactor for Tat-mediated transactivation of HIV-1 (Wei et al, 1998). To initiate HIV-1 replication, P-TEFb is recruited by Tat to the HIV-1 long terminal repeat (LTR), where P-TEFb phosphorylates host RNA polymerase II (RNA Pol II), thereby promoting highly efficient transcription and leading to the production of full-length viral transcripts (Mousseau and Valente, 2017). Therefore, the progression of HIV-1 within infected cells is determined by the activity of Tat (Donahue et al, 2012). Specifically speaking, Tat initiates a positive feedback loop that greatly promotes HIV-1 replication, whereas when this loop is halted, HIV-1 may enter dormancy, which could be reversed by introducing Tat or other transcription activator such as NF-κB and SP1 (Jordan et al, 2003; Razooky et al, 2015).

Typically, more than half of P-TEFb is sequestered by the 7SK small nuclear ribonucleoprotein (7SK snRNP), a large ribonucleoprotein complex assembly that impedes the kinase activity of P-TEFb, thereby preventing phosphorylation of RNA Pol II (Nguyen et al, 2001). The 7SK snRNP consists of 7SK small nuclear RNA (7SK snRNA) along with several interacting proteins, including La-related protein 7 (LARP7), methylphosphate capping enzyme (MePCE) and a homodimer or heterodimer of hexamethylene bisacetamide-inducible proteins 1 and 2 (HEXIM1 and HEXIM2) (C Quaresma et al, 2016). LARP7, a member of the La-related protein family, is instrumental in stabilizing the 7SK snRNP complex and directly inhibits P-TEFb activity (Krueger et al, 2008). Emerging evidence suggests that LARP7 also exerts an inhibitory effect on HIV-1 replication (Markert et al, 2008). Therefore, elucidating how LARP7 regulates P-TEFb and how P-TEFb is subsequently hijacked by Tat is critical for developing strategies to inhibit HIV-1 replication (Barboric et al, 2007; Sobhian et al, 2010).

Liquid–liquid phase separation (LLPS) has been found to participate in multiple cellular and viral processes, including gene transcription, RNA metabolism and virus assembly (Boija et al, 2018; Lafontaine, 2019; Lu et al, 2018; Wei et al, 2022). Proteins that undergo LLPS frequently contain intrinsically disordered regions (IDRs) (Hyman et al, 2014). These proteins, along with other biomolecules, undergo phase separation to form distinct, membrane-less organelles, thereby contributing to the compartmentalization of specific biological functions within the cell (Hirose et al, 2023; Pappu et al, 2023). Both the transcription initiation and

[1]Nanhu Laboratory, National Center of Biomedical Analysis, Beijing 100039, China. [2]Institute of Translational Medicine, Zhejiang University, Hangzhou 310029, China. [3]School of Basic Medical Sciences, Fudan University, Shanghai 200032, China. ✉E-mail: tnli@ncba.ac.cn; whli@ncba.ac.cn

splicing machineries, containing large numbers of component molecules, have been reported to be condensates formed by phase separation (Guo et al, 2019). The involvement of LLPS in viral infections has garnered considerable recent attention (Li et al, 2022). For example, the N protein of SARS-COV-2 has been demonstrated to facilitate viral replication and assembly through LLPS (Iserman et al, 2020; Zhao et al, 2021). In the case of HIV-1, nucleocapsid proteins regulate genomic RNA positioning and trafficking through zinc finger protein-dependent LLPS (Monette et al, 2020). Additionally, the host protein CPSF6 forms co-condensates with the HIV-1 capsid, serving as reverse transcription hubs to generate new viral DNA (Bejarano et al, 2019; Luchsinger et al, 2023). These mechanisms represent critical steps in the HIV-1 replication cycle. Consequently, investigating the LLPS of both viral and host proteins during HIV-1 infection is of great interest for understanding the mechanisms of HIV-1 replication and host cell resistance.

In this study, we discovered that HIV-1 infection enhances the LLPS of LARP7. We observed that LARP7 compartmentalizes and concentrates P-TEFb and Tat within LLPS droplets, inhibiting Tat-mediated HIV-1 transcription. Furthermore, our study demonstrates that conserved lysine residues within the IDR of LARP7 are essential for its LLPS. Moreover, LARP7 mutants deficient in phase separation lose their ability to inhibit Tat-mediated transcription. These results highlight the crucial role of LARP7 LLPS in the inhibition of Tat-mediated HIV-1 infection. Collectively, our findings offer novel insights into the mechanisms underlying host defense against HIV-1 infection through LLPS.

## Results and discussion

### HIV-1 infection promotes the liquid–liquid phase separation of LARP7

In our quest to identify host proteins involved in HIV-1 infection that might function through phase separation, we analyzed– 64 proteins documented in HIV-1 research literature (see Table 1) using PONDR (http://pondr.com/), a predictor of IDRs, which are prevalent in proteins that undergo LLPS (Hyman et al, 2014; Ray and Maji, 2020). We found that LARP7 ranked seventh in this analysis, with scores of 0.64 (VL3), 0.67 (VSL2), and 0.58 (VL-XL) from the three PONDR algorithms; a score > 0.5 indicates a high likelihood of LLPS. The top 20 proteins are shown in Fig. EV1A. In addition, LARP7 placed highly in the results from another predictor, IUPred2A (https://iupred2a.elte.hu/), with proteins scoring > 0.5 shown in Fig. EV1B.

Subsequently, we used immunofluorescence to examine the distribution of LARP7 in Jurkat cells infected with an HIV-1 pseudovirus (R7-ΔEnv-GFP, where the *GFP* gene is inserted into the HIV-1 *env* gene) to assess whether LARP7 undergoes LLPS upon HIV-1 infection (Campbell et al, 2007). In uninfected cells, LARP7 was predominantly diffused with minor concentrated puncta present in the nucleus. Following HIV-1 infection, LARP7 puncta exhibited a significant increase in both number and fluorescence intensity. These droplets persisted for approximately 24 h post-infection and then began to transition to a more homogeneous morphology (Fig. 1A,B). The observed change in the LARP7 staining pattern in Jurkat cells infected with HIV-1 was not attributable to alter LARP7 expression because western blotting

**Table 1. Prediction results of the phase separation propensity for host cell proteins associated with HIV-1 infection.**

| | | Predictors | | | |
|---|---|---|---|---|---|
| | | **Pondr** | | | **IUPRED2A** |
| | **Proteins** | **VL3** | **VSL2** | **VL-XT** | **IUPRED2** |
| 1 | CD4 | 0.41 | 0.42 | 0.30 | 0.24 |
| 2 | CXCR4 | 0.28 | 0.27 | 0.16 | 0.06 |
| 3 | CCR5 | 0.16 | 0.18 | 0.11 | 0.03 |
| 4 | CD209 | 0.40 | 0.53 | 0.32 | 0.32 |
| 5 | CYPA | 0.25 | 0.37 | 0.20 | 0.27 |
| 6 | PIN1 | 0.58 | 0.68 | 0.44 | 0.50 |
| 7 | NUP358 | 0.48 | 0.57 | 0.32 | 0.37 |
| 8 | NUP153 | 0.67 | 0.82 | 0.38 | 0.50 |
| 9 | NUP98 | 0.47 | 0.60 | 0.33 | 0.40 |
| 10 | Importin-7 | 0.27 | 0.30 | 0.25 | 0.20 |
| 11 | UNG | 0.35 | 0.47 | 0.31 | 0.37 |
| 12 | DCAF1 | 0.39 | 0.45 | 0.37 | 0.31 |
| 13 | LEDGF | 0.71 | 0.79 | 0.64 | 0.71 |
| 14 | Transportin-3 | 0.20 | 0.25 | 0.20 | 0.11 |
| 15 | SMARCB1 | 0.39 | 0.43 | 0.41 | 0.35 |
| 16 | LARP7 | 0.64 | 0.67 | 0.58 | 0.50 |
| 17 | Cyclin T1 | 0.56 | 0.65 | 0.45 | 0.53 |
| 18 | CDK9 | 0.29 | 0.36 | 0.23 | 0.20 |
| 19 | DDX3 | 0.41 | 0.51 | 0.38 | 0.36 |
| 20 | NAP1 | 0.24 | 0.30 | 0.26 | 0.15 |
| 21 | P300 | 0.68 | 0.75 | 0.65 | 0.63 |
| 22 | CBP | 0.67 | 0.74 | 0.65 | 0.62 |
| 23 | Importin β1 | 0.28 | 0.32 | 0.31 | 0.21 |
| 24 | ICRM1 | 0.19 | 0.27 | 0.20 | 0.12 |
| 25 | B23 | 0.61 | 0.68 | 0.58 | 0.57 |
| 26 | DDX1 | 0.25 | 0.35 | 0.23 | 0.26 |
| 27 | DDX5 | 0.33 | 0.45 | 0.39 | 0.36 |
| 28 | TSG101 | 0.44 | 0.54 | 0.49 | 0.35 |
| 29 | ALIX | 0.42 | 0.50 | 0.45 | 0.36 |
| 30 | Calmodulin | 0.51 | 0.55 | 0.46 | 0.46 |
| 31 | DHX9 | 0.37 | 0.41 | 0.34 | 0.28 |
| 32 | STAU1 | 0.60 | 0.65 | 0.50 | 0.52 |
| 33 | EAP30 | 0.27 | 0.35 | 0.28 | 0.10 |
| 34 | ABCE1 | 0.29 | 0.33 | 0.28 | 0.18 |
| 35 | BST2 | 0.43 | 0.44 | 0.38 | 0.22 |
| 36 | BTRC | 0.54 | 0.61 | 0.36 | 0.37 |
| 37 | AP1M1 | 0.25 | 0.33 | 0.25 | 0.19 |
| 38 | ABCA1 | 0.26 | 0.33 | 0.23 | 0.19 |
| 39 | Calnexin | 0.54 | 0.56 | 0.45 | 0.48 |
| 40 | ACOT8 | 0.34 | 0.40 | 0.26 | 0.25 |

**Table 1.** (continued)

| | | Predictors | | | |
|---|---|---|---|---|---|
| | | Pondr | | | IUPRED2A |
| | Proteins | VL3 | VSL2 | VL-XT | IUPRED2 |
| 41 | PACS-1 | 0.55 | 0.59 | 0.50 | 0.46 |
| 42 | PACS-2 | 0.55 | 0.59 | 0.49 | 0.46 |
| 43 | PAK2 | 0.50 | 0.52 | 0.40 | 0.41 |
| 44 | LCK | 0.27 | 0.34 | 0.28 | 0.26 |
| 45 | HCK | 0.35 | 0.43 | 0.31 | 0.30 |
| 46 | APOBEC3G | 0.13 | 0.25 | 0.12 | 0.10 |
| 47 | APOBEC3F | 0.10 | 0.23 | 0.15 | 0.06 |
| 48 | ELOB | 0.58 | 0.60 | 0.50 | 0.47 |
| 49 | CUL5 | 0.26 | 0.35 | 0.26 | 0.18 |
| 50 | ELOC | 0.25 | 0.37 | 0.39 | 0.26 |
| 51 | SAHMD1 | 0.25 | 0.33 | 0.25 | 0.22 |
| 52 | TRIM5aα | 0.37 | 0.40 | 0.31 | 0.21 |
| 53 | CPSF6 | 0.73 | 0.78 | 0.64 | 0.72 |
| 54 | NUP62 | 0.62 | 0.76 | 0.53 | 0.48 |
| 55 | Importin α1 | 0.33 | 0.38 | 0.30 | 0.23 |
| 56 | MED14 | 0.34 | 0.42 | 0.37 | 0.30 |
| 57 | TIP110 | 0.46 | 0.51 | 0.44 | 0.36 |
| 58 | SAM68 | 0.74 | 0.74 | 0.57 | 0.64 |
| 59 | DDX24 | 0.51 | 0.56 | 0.46 | 0.46 |
| 60 | RBX1 | 0.24 | 0.31 | 0.26 | 0.15 |
| 61 | NFAT1 | 0.61 | 0.65 | 0.60 | 0.55 |
| 62 | RAD23A | 0.61 | 0.65 | 0.62 | 0.58 |
| 63 | Catenin β1 | 0.31 | 0.39 | 0.36 | 0.32 |
| 64 | GEMIN2 | 0.44 | 0.47 | 0.42 | 0.30 |

analysis confirmed that LARP7 expression remained constant regardless of the HIV-1 infection status (Figs. 1C and EV1D). The specificity of a LARP7 antibody was validated through immuno-fluorescence and immunoblotting experiments in Jurkat cells with stable LARP7 knockdown (Fig. EV1C). We confirmed this phenomenon in primary CD4[+] T cells isolated from primary peripheral blood mononuclear cells (PBMCs) of healthy individuals (Fig. EV1E). In these cells, after HIV-1 infection, the formation of LARP7 condensates was significantly enhanced, with both the number and fluorescence intensity of puncta markedly increasing, while the overall expression level of LARP7 remained unchanged (Figs. 1D–F and EV1F). Similar experiments were conducted in Tzm-bl cells, which is an engineered HeLa cell line expressing CD4 and CCR5 and Tat-activated HIV-LTR luciferase reporter system, and is prevalently used to study HIV-1 infection and Tat-mediated transactivation (Montefiori, 2009). After Tzm-bl cells were infected with HIV-1 for 24 h, a significantly higher number of LARP7 puncta can be observed in virus-infected cells expressing GFP compared to GFP-negative cells, while LARP7 expression level was the same in these cells (Fig. EV1G–I).

To test whether HIV-1 infection induced the formation of LARP7 condensates through LLPS, we treated Jurkat cells infected by HIV-1 for 12 h with 1,6-hexanediol (1,6-HD), a broad-spectrum inhibitor of LLPS, while employing 2,5-hexanediol (2,5-HD) as a negative control (Lin et al, 2016). Observations revealed a substantial decrease in LARP7 puncta in cells treated with 1,6-HD, but not in cells treated with 2,5-HD, thereby confirming that these puncta are formed through LLPS (Fig. 1G,H).

Interestingly, neither Herpes simplex virus (HSV-GFP) nor Vesicular stomatitis virus (VSV-GFP) infection induced the formation of LARP7 puncta in Jurkat cells, suggesting that the induction of LARP7 condensates by viral infection was specific to HIV-1 (Fig. 1I,J).

To validate the phase separation of LARP7, we overexpressed mCherry-LARP7 in Tzm-bl cells and monitored the dynamics of LARP7 droplets. As shown in Movie EV1, active LARP7 droplet fusion could be observed (Fig. EV2A). To further investigate the phase separation features of LARP7, we adopted a photoinduced phase separation research system (OptoIDR system). Cry2 is a light-sensitive protein which is known to self-associate upon blue light exposure. When fused with a phase-separating protein, specifically an IDRs containing protein or fragment, Cry2 will mediate blue light-dependent phase separation of the fusion protein in the cell (Shin et al, 2017) (Fig. 1K). As shown in Fig. 1L, Cry2-mCherry-LARP7 formed spherical droplets under blue light, and droplet fusion was observed, exhibiting typical liquid-like properties. By contrast, Cry2-mCherry alone did not form droplets after blue light stimulation (Fig. EV2B). These results suggest that the formation of the Cry2-mCherry-LARP7 droplets was driven by the phase separation of LARP7. Additionally, fluorescence recovery after photobleaching (FRAP) analysis indicated that these droplets undergo dynamic exchange with the surrounding dilute phase (Fig. 1M,N).

We also verified LARP7 phase separation in vitro. mCherry-LARP7 protein purified from SF9 cells was diluted to various concentrations in a phase separation buffer containing 5% polyethylene glycol (PEG)-3000 as a crowding agent. Spherical droplets were observed, with both the size and number of droplets increasing with the LARP7 concentration (Fig. EV2C). The partition coefficient, calculated by dividing the total fluorescence intensity within the droplets by the background fluorescence intensity, represents the degree of protein phase separation in solution (Banani et al, 2016). We observed an increase in the partition coefficient with increasing LARP7 concentration, indicating concentration-dependent phase separation of LARP7 in vitro (Fig. EV2D). These droplets disappeared upon increasing the NaCl concentration to 500 mM or adding 6% 1,6-HD to the solution (Fig. EV2E–H), supporting the conclusion that LARP7 condensates are formed through LLPS.

Taken together, our results suggest that phase separation of LARP7 is induced when cells are infected by HIV-1, and LARP7 undergoes LLPS both *in cellulo* and in vitro.

## Tat is incorporated into LARP7 condensates upon HIV-1 infection

LARP7, a core component of the 7SK snRNP complex, stabilizes the complex, enabling 7SK snRNP to bind to P-TEFb and maintain it in an inactive state, thereby inhibiting transcription (Krueger et al, 2008). During HIV-1 infection, the viral transactivator Tat hijacks P-TEFb from the 7SK snRNP complex, initiating

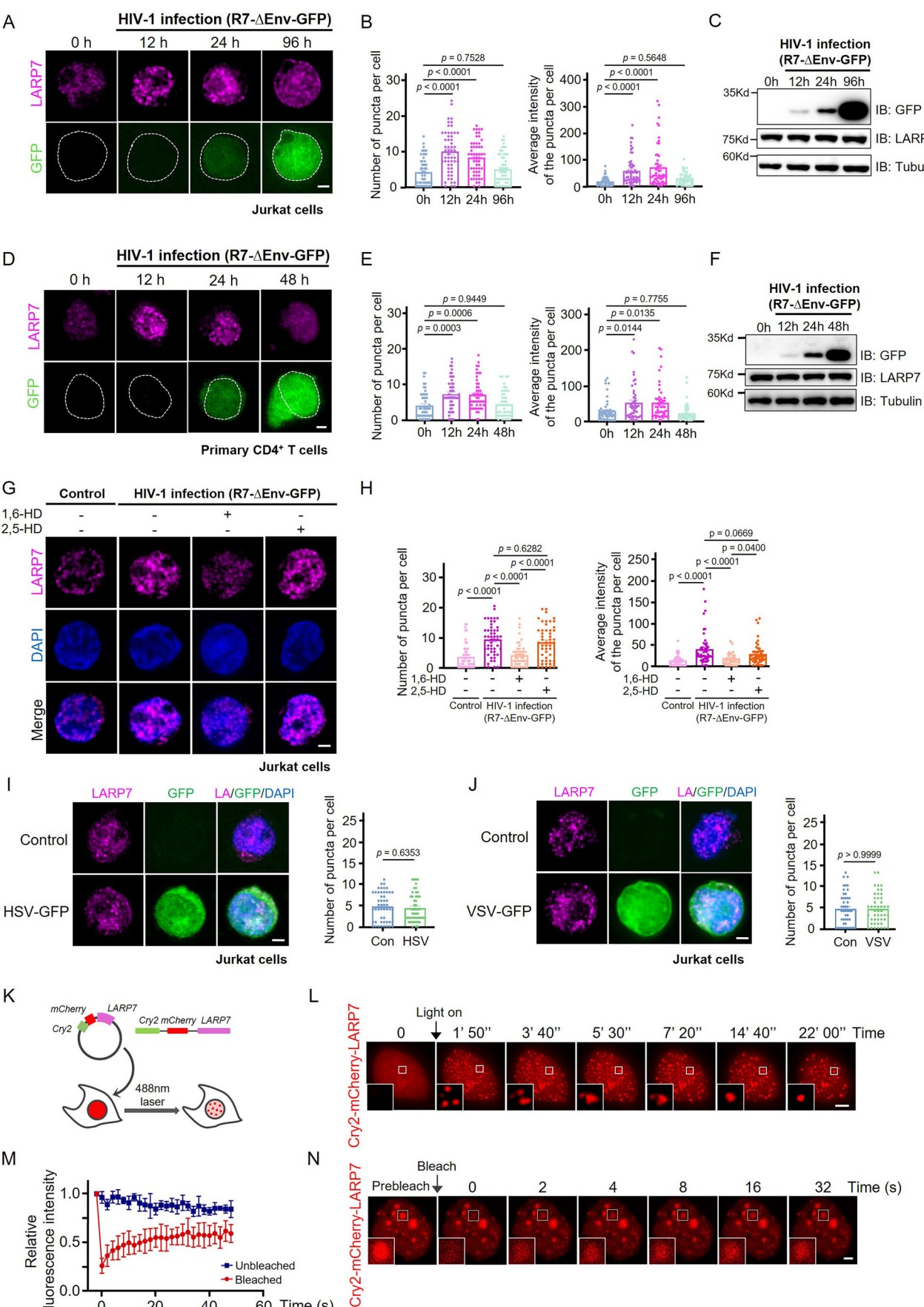

**Figure 1. HIV-1 infection promotes the liquid–liquid phase separation of LARP7.**

(A) Representative images of the immunofluorescence of LARP7 in Jurkat cells at different times after infection by HIV-1 R7-ΔEnv-GFP, the white dotted lines indicate the nucleus. Scale bar, 2 μm. (B) Quantification of the number (left) and the average intensity (right) of puncta from individual cells in Fig. 1A. Fifty cells were analyzed in each group. One-way analysis of variance (ANOVA) was used for statistical analysis, and exact $p$ values are represented in the figure, mean ± SEM. (C) Western blotting analysis of LARP7 expression in Jurkat cells at different times after HIV-1 R7-ΔEnv-GFP infection in Fig. 1A. (D) Representative images of the immunofluorescence of LARP7 in primary CD4$^+$ T cells at different times after infection by HIV-1 R7-ΔEnv-GFP, the white dotted lines indicate the nucleus. Scale bar, 2 μm. (E) Quantification of the number (left) and the average intensity (right) of puncta from individual cells in Fig. 1D. Fifty cells were analyzed in each group. One-way analysis of variance (ANOVA) was used for statistical analysis, and exact $p$ values are represented in the figure, mean ± SEM. (F) Western blotting analysis of LARP7 expression in primary CD4$^+$ T cells at different times after HIV-1 R7-ΔEnv-GFP infection in Fig. 1D. (G) Representative images of the immunofluorescence of LARP7 in Jurkat cells infected by HIV-1 R7-ΔEnv-GFP. After infection for 12 h, cells were treated with 6% 1,6-hexanediol (1,6-HD) or 2,5-hexanediol (2,5-HD) for 30 s before fixing. Scale bar, 2 μm. (H) Quantification of the number (left) and the average intensity (right) of puncta from individual cells treated as in Fig. 1G. Fifty cells were analyzed in each group. One-way analysis of variance (ANOVA) was used for statistical analysis, and exact $p$ values are represented in the figure, mean ± SEM. (I) Representative images of the immunofluorescence of LARP7 in Jurkat cells 24 h after HSV-GFP infection. Scale bar, 2 μm (left). Quantification of the number of puncta from individual cells. Fifty cells were analyzed in each group. A two-tailed unpaired student's $t$-test was used for statistical analysis and exact $p$ values are represented in the figure, mean ± SEM (right). (J) Representative images of the immunofluorescence of LARP7 in Jurkat cells 24 h after VSV-GFP infection. Scale bar, 2 μm (left). Quantification of the number of puncta from individual cells. Forty-seven cells were analyzed in each group. A two-tailed unpaired student's $t$-test was used for statistical analysis and exact $p$ values are represented in the figure, mean ± SEM (right). (K) Schematic of the photoinduced phase separation assay. Fused recombinant Cry2 (green), mCherry (red), and LARP7 (purple) was expressed in cells and then exposed to blue light (488-nm laser). (L) Timelapse imaging of cells expressing Cry2-mCherry-LARP7 subjected to laser excitation every 1 min 50 s. Fusion of Cry2-mCherry-LARP7 droplets occurs in the region enlarged in the white box. Scale bar, 5 μm. (M) Quantitative analysis of the average recovery rate of Cry2-mCherry-LARP7 puncta fluorescence in Fig. 1N. Each data point represents an independent biological replicate ($n = 3$), mean ± SD. (N) Representative images of fluorescence recovery after photobleaching (FRAP) experiments with Cry2-mCherry-LARP7-expressing cells; the bleached region is shown enlarged in the white box. Scale bar, 2 μm. Source data are available online for this figure.

transcription of HIV-1. Thus, we sought to investigate whether the LARP7 condensates induced by HIV-1 infection are related to its transcriptional inhibition function. Immunofluorescence staining was employed to investigate the presence of another 7SK snRNP complex component, HEXIM1 and P-TEFb component in HIV-1-induced LARP7 condensates. As shown in Fig. 2A,B, HEXIM1 and Cyclin T1 co-concentrated in LARP7 puncta after HIV-1 infection. Additionally, Tat foci were observed to co-localize with LARP7, suggesting that Tat also enters HIV-1-induced LARP7 condensates (Fig. 2C,D). These co-localized puncta disappeared following treatment with 1,6-HD, indicating that LLPS-driven LARP7 droplets incorporate with Tat during HIV-1 infection.

Next, we further investigated the possibility that LARP7 and Tat co-condense through LLPS. mCherry-LARP7 and mEGFP-Tat formed co-localized condensates when they were co-expressed in HEK293T cells. The fluorescence of both LARP7 and Tat within the co-condensates area recovered rapidly after photobleaching, indicative of the liquidity of LARP7 and Tat condensates and active molecular exchange between the condensate phase and dilute phase (Fig. 2E).

Interestingly, the morphology of the overexpressed mCherry-LARP7 and mEGFP-Tat looked like nucleoli, which is different from the immunofluorescence staining result. We performed nucleolus staining using an anti-NPM1(B23) antibody while investigating the localization of LARP7 and Tat. As shown in Fig. EV3A, mCherry-LARP7 co-condensates with mEGFP-Tat in nucleoli. Though it looks very different, LARP7 puncta can be easily found residing in the nucleolus of HIV-1 infected CD4$^+$ T cells (Fig. EV3B). It is a consensus that the nucleolus is also an LLPS structure (Lafontaine et al, 2021), the co-condensates of LARP7 and Tat in nucleolus might be corelated with their LLPS property. The expression level and the different composition of interacting molecules, including proteins and RNAs in different cells, might be the reason that LARP7 concentration in nucleolus is much more pronounced in transfected HEK293T cells.

To exclude the influence of other proteins and RNAs in cells, we purified the mCherry-LARP7 and mEGFP-Tat proteins in SF9 cells

and investigated their co-phase separation in vitro. We found that purified mEGFP-Tat forms bead-like precipitates in phase separation buffer when reaching a certain concentration. Interestingly, upon the addition of LARP7, the two proteins co-phase-separated. What shape of the co-condensates would be, liquid droplets or solid particles, is dependent on the concentrations as well as the relative proportion of the two proteins in the solution (Fig. 2F). As the relative proportion of LARP7 increased, the two proteins tend to co-condensate as spherical droplets, otherwise, as irregularly shaped aggregates. Figure 2G presents a phase diagram illustrating LARP7 and Tat co-condensates at varying protein concentrations. These findings suggest that a higher LARP7/Tat ratio enhances their co-liquid-liquid phase separation, prompting further investigation into its impact on HIV-1 transcription. We transiently knocked down LARP7 in Tzm-bl cells and subsequently expressed varying levels of Tat, the cells were then infected with HIV-1 for 24 h and assessed for viral replication (Fig. EV3C,D). The results indicated that when LARP7 was present, the pre-expressed Tat cannot further enhance HIV-1 transcription. In contrast, when LARP7 was absent, there was a significant increase in HIV-1 transcription, and it was further enhanced with the rise in the amount of pre-expressed Tat. We observed that under certain conditions, a higher LARP7/Tat ratio correlates with reduced HIV-1 transcription. These findings suggested that LARP7 plays a crucial role in regulating Tat-mediated HIV-1 transcription. Collectively, our results indicate that LARP7 and Tat co-phase-separate both in cellulo and in vitro, with LARP7 phase separation potentially playing a crucial role in regulating Tat-mediated HIV-1 transcription.

## Conserved lysine residues in the IDR are essential for the phase separation of LARP7

Next, we aimed to identify the structural features of LARP7 that are critical for its LLPS. Using the PONDR predictor, we analyzed the amino acid sequence of LARP7, aiming to identify its IDRs. The

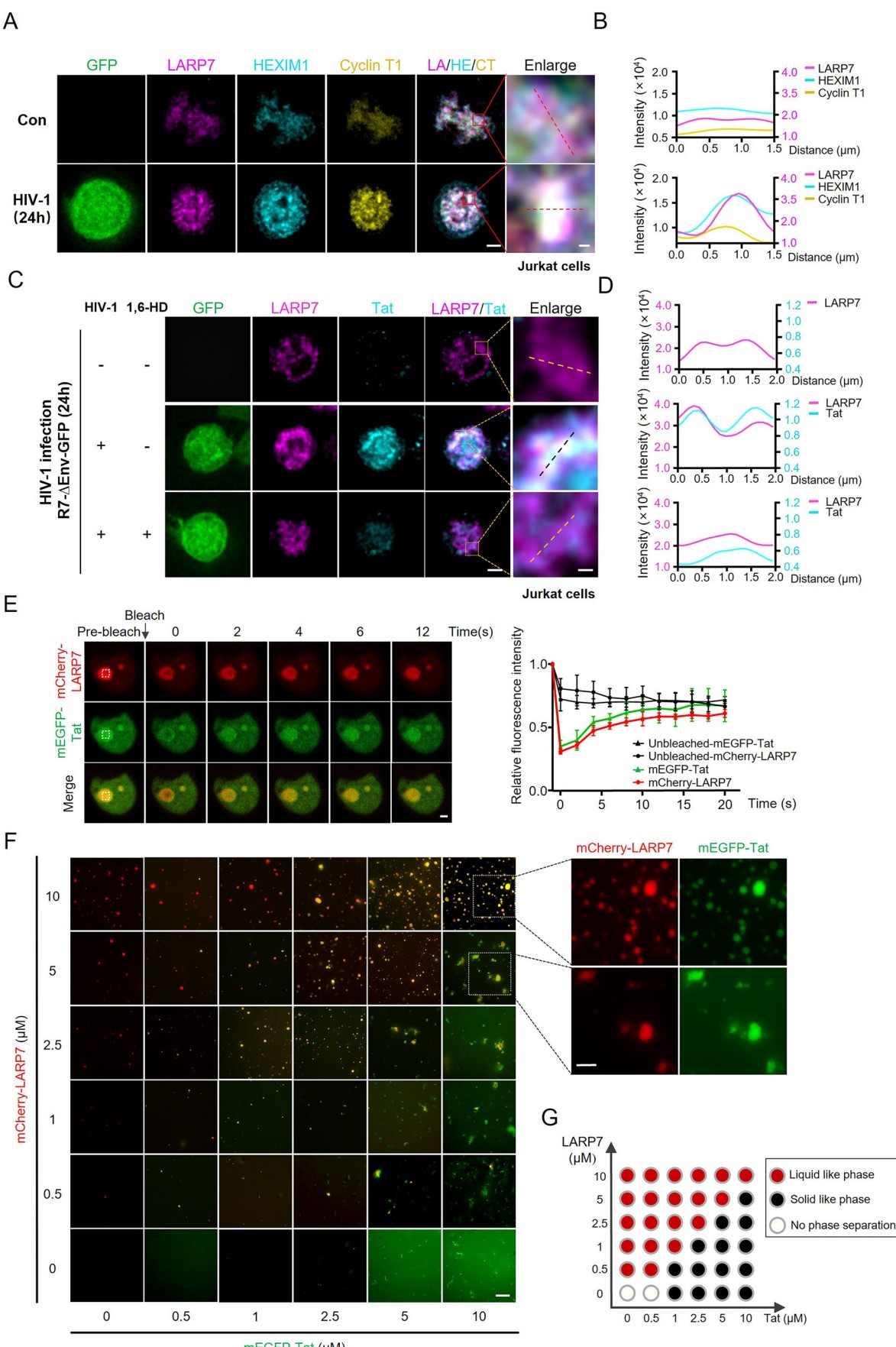

**Figure 2. Tat is incorporated into LARP7 condensates upon HIV-1 infection.**

(A) Representative images of the immunofluorescence of LARP7, HEXIM1, and Cyclin T1 in Jurkat cells 24 h after infection by HIV-1 R7-ΔEnv-GFP. Scale bar, 2 μm. Co-localized droplets are shown in the enlarged region. Scale bar, 0.5 μm. The fluorescence intensity profiles along the red dotted lines are shown in Fig. 2B. (B) Fluorescence intensity profiles of LARP7 (magenta curve), HEXIM1(cyan curve,) and Cyclin T1 (yellow curve) on the red dotted lines indicated in the images in Fig. 2A. (C) Representative images of the immunofluorescence of LARP7 and Tat in Jurkat cells 24 h after infection by HIV-1 R7-ΔEnv-GFP. Cells were treated with 6% 1,6-HD for 30 s before fixing. Scale bar, 2 μm. Co-localized droplets are shown in the enlarged region. Scale bar, 0.5 μm. The fluorescence intensity profiles along the yellow and black dotted lines are shown in Fig. 2D. (D) Fluorescence intensity profiles of LARP7 (magenta curve) and Tat (cyan curve) on the dotted lines indicated in the images in Fig. 2C. (E) Left, FRAP experiment on cells co-expressing mCherry-LARP7 and mEGFP-Tat. Scale bar, 5 μm. Right, quantitative analysis of the average fluorescence recovery rate of mCherry-LARP7 (red curve) and mEGFP-Tat (green curve) puncta. Each data point represents an independent biological replicate ($n = 3$), mean ± SD. (F) Representative images of mixtures of purified mCherry-LARP7 and mEGFP-Tat proteins with the indicated concentrations. Scale bar, 20 μm. Co-localized droplets are shown in the enlarged region. Scale bar, 10 μm. (G) Phase diagrams of mCherry-LARP7 and mEGFP-Tat mixtures with concentrations ranging from 0.5 to 10 μM. Source data are available online for this figure.

results revealed a long, continuous IDR between residues 189 and 449 (Fig. 3A). IDRs are characterized by their lack of a stable three-dimensional structure, can have a biased amino acid composition, and frequently exhibit repetitive sequence motifs. Notably, the IDR of LARP7 is enriched with lysine residues, with 51 of the 261 amino acids being lysine. The positive charge on these lysine residues is hypothesized to play a pivotal role in LLPS (Fig. 3B) (Wang et al, 2021). Furthermore, these lysine residues are evolutionarily conserved (Fig. 3C). Therefore, we speculated that the lysine residues in the LARP7 IDR play an important role in its phase separation.

To investigate the role of these lysine residues in the LARP7 IDR in LLPS, variants were constructed by mutation of selected lysine residues to alanine. We first mutated 49 of the 51 lysine residues in LARP7 IDR simultaneously to alanine (retaining two lysine residues within the nuclear localization sequence), and then we reduced the mutated lysine residues and constructed the mutant 30KA (mutated the most contiguous and conserved 30 lysine residues), 21KA (recover lest conversed lysine residues from 30KA in 195aa–230aa and 274–314aa) and 13KA (only the most contiguous lysine residues were mutated, 218–225aa and 274–278aa) (Fig. 3D). Droplets formation, subcellular localization and co-condensation with Tat of these mutants were observed through OptoIDR system and overexpression fluorescence-fused protein in cells. As shown in Figs. 3E and EV3E-G, 49KA mutant completely abolished droplet formation, nucleolus localization and the ability to co-condensate with Tat. 30KA mutant showed similar behavior in OptoIDR system with 49KA mutant (Fig. 3E), and exhibited no obvious co-condensation with Tat in nucleolus (Fig. EV3F,G). 21KA and 13KA mutants exhibited distinct restoration of phase separation and co-condensate with Tat. These results suggest that the lysine residues in IDR are important for LARP7 to undergo LLPS and co-condensate with Tat in cells.

Our findings imply that the phase separation of LARP7 depends on the lysine residues in its IDR, and these lysine residues also influence the distribution of LARP7 in the nucleus.

## LARP7 inhibits Tat-mediated HIV-1 infection through phase separation

To explore the function of LARP7 phase separation in Tat-dependent activation of transcription, we transiently knocked down LARP7 in Tzm-bl cells, stably expressing different LARP7 mutants. These cells were then transfected with Tat and a *Renilla* luciferase

reporter plasmid (pRL-TK), and the level of HIV-1-LTR luciferase activity was assessed using a dual-luciferase reporter assay (Fig. EV4A). Transient knockdown of LARP7 led to increase relative luciferase activity, indicating that LARP7 represses Tat-mediated transcription; this was rescued by expression of wild-type LARP7, but not by the LLPS-deficient 49KA and 30KA mutants. The inhibitory effect was restored in the 21KA and 13KA mutants, along with their phase separation ability (Figs. 4A and EV4B). We also transiently knocked down LARP7 in Tzm-bl cells stably expressing different LARP7 mutants, infected the cells with HIV-1 R7-ΔEnv-GFP, and detected GFP expression 48 h after infection using FACS and western blotting to assess the level of HIV-1 transcription. The results were consistent with those of the luciferase activity assays, confirming that phase separation of LARP7 is important for the inhibition of Tat-mediated HIV-1 transcription (Figs. 4B,C and EV4C). Consistent findings were made when measuring HIV-1 mRNA (*ltr* and *pol*) expression post-infection (Fig. 4D).

Since binding with 7SK snRNA, P-TEFb and the protein components of 7SK snRNP, such as HEXIM1 and MePCE are believed to be essential for LARP7 to regulate transcription. We subsequently evaluated the binding affinity of LARP7 mutants to 7SK snRNA by using RNA immunoprecipitation (RIP)-qPCR. The results showed that the 49KA mutant almost completely lost the binding of 7SK snRNA. Although the 30KA mutant did not undergo phase separation, it retained its ability to associate with 7SK snRNA (Fig. 4E). We further investigated the interactions between LARP7 mutants and P-TEFb components, as well as HEXIM1 and MePCE through immunoprecipitation assays. The results revealed that 49KA mutation almost destroyed the binding of the 49-lysine mutant to P-TEFb, HEXIM1 and MePCE, while significantly reduced for 30KA mutant, a gradual restoration of LARP7 interaction with P-TEFb and HEXIM1 was observed as the mutations were recovered in the 30KA, 21KA, and 13KA mutants (Fig. 4F). These findings indicated that the mutation of lysine residues in the IDR of LARP7 not only disrupts the phase separation of this protein but also impairs its interaction with P-TEFb and one 7SK snRNP component HEXIM1. Interestingly, the impact of lysine mutations on LARP7 interaction with MePCE is similar to that on LARP7 interaction with 7SK snRNA. Specifically speaking, a non-phase separated mutant, 30KA, bound to a similar amount of MePCE with wild-type LARP7. These data suggest that LARP7 LLPS is important for its interaction with P-TEFb and HEXIM1, but not required for its interaction with MePCE and 7SK snRNA. On the other hand, interactions

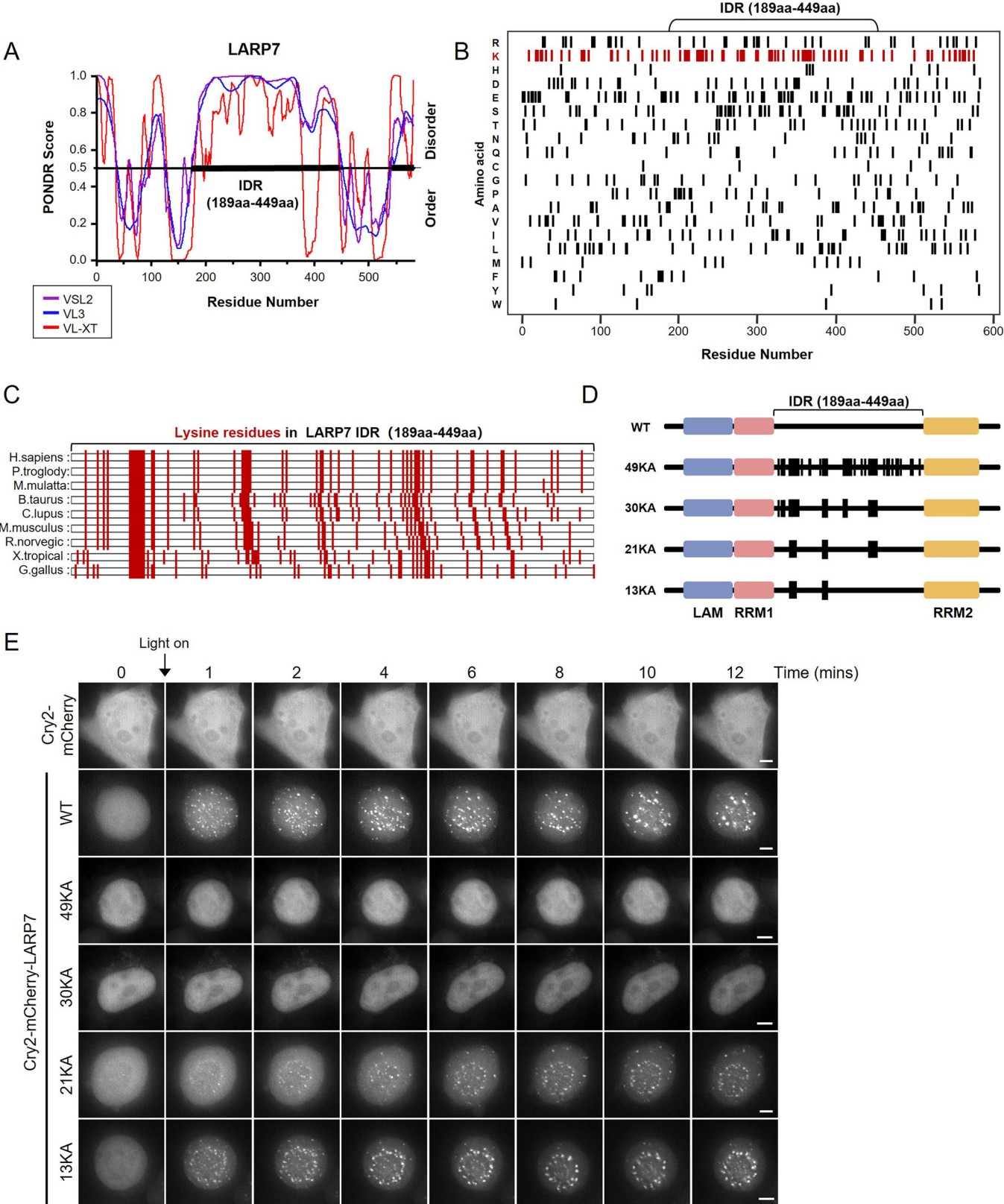

**Figure 3. Conserved lysine residues in the intrinsically disordered region (IDR) are essential for the phase separation of LARP7.**

(A) IDRs of LARP7 predicted by using PONDR. The disorder tendency is scored between 0 and 1, and scores > 0.5 indicate disorder. (B) Amino acid composition of LARP7. Each row represents a particular amino acid; black bars represent the presence of the indicated amino acid at that position. (C) Lysine composition of LARP7 (189aa–449aa) from different species; data were analyzed by using MEGA 7.0 software. (D) Domain structure of LARP7 and schematic of wild-type (WT) LARP7 and mutants with different numbers of lysine residues changed to alanine (49KA, 30KA, 21KA, and 13KA). (E) Timelapse imaging of cells expressing Cry2-mCherry-LARP7(WT/mut). Scale bar, 5 μm. Source data are available online for this figure.

with MePCE and 7SK snRNA is not sufficient for LARP7 LLPS and its function on regulating Tat mediate transactivation.

In the absence of a cure for AIDS, it is crucial to search for effective ways to regulate Tat-induced HIV-1 transcription. Our findings provide evidence that the phase separation of LARP7 can be specifically promoted by HIV-1 infection, leading to the formation of 7SK snRNP condensates that concentrate P-TEFb and Tat, assembling an inhibitory regulator of viral transcription elongation. We demonstrated the LLPS of LARP7 both in isolation and in conjunction with Tat, *in cellulo* and in vitro. Interestingly, while Tat alone formed bead-like precipitates in vitro, the presence of LARP7 facilitated the formation of fluid droplets, indicating that LARP7 and Tat co-condense through LLPS at specific protein ratios. Notably, mutation of lysine residues in the LARP7 IDR disrupted these condensates and abolished the inhibitory effect on Tat-mediated transcription activation, suggesting that the LLPS of LARP7 is essential for restricting Tat-mediated HIV-1 replication. Considering all the results above, a working model of inhibition of HIV-1 infection by LARP7 LLPS can be proposed. In the resting state, part of the P-TEFb is imprisoned in membrane-less organelles formed by LARP7, which controls its activity (Fig. EV4D, left). In the early stage of HIV-1 infection, Tat starts to be expressed, but it is unable to recruit P-TEFb to initiate HIV-1 LTR transcription because the relatively low levels of Tat are trapped together with P-TEFb within the LARP7-induced droplets (Fig. EV4D, right).

Recent insights into the role of phase separation in viral infection have highlighted its significance in optimizing viral replication by concentrating replicase proteins, viral genomes, and host factors necessary for viral replication, while potentially shielding viruses from innate immune responses (Charman et al, 2023; York, 2021). In this study, we have found a novel mechanism of host resistance to HIV-1 infection: by forming macromolecular condensates through LLPS, host protein LARP7 confines the HIV-1 transactivator Tat upon viral entry. However, the precise mechanisms of how and when LARP7 detects the infection remain subjects for further research.

HIV-1 Tat plays a crucial role in viral replication by sustaining high levels of transcription throughout the viral life cycle. Tat recruits P-TEFb, a kinase complex that phosphorylates the *C*-terminal domain of RNA Pol II, to the HIV-1 LTR through binding to the transactivation response (TAR) hairpin, thereby promoting transcription elongation (Lu et al, 2013; Sedore et al, 2007). The subnuclear localization of Tat is recognized as critical for its function because it frequently shuttles between the nucleolus and the nucleoplasm. It has been reported that when localized in the nucleolus, Tat appears to exist in a state that inhibits transcriptional activation (Jin et al, 2023). The co-localization of LARP7 and Tat in the nucleolus may contribute to the maintenance of this inhibited state; nevertheless, the underlying mechanism warrants further investigation.

Latently infected cells present a significant challenge to the eradication of HIV-1 (Margolis et al, 2016). Tat-dependent

transcription is known to be the "switch" for reactivating latent HIV-1 (Donahue et al, 2012). We showed here that LARP7 LLPS, induced by HIV-1 infection, restrains Tat-dependent transcription, which could serve as a strategy for the host to resist viral infection and, conversely, act as a mechanism driving viral latency. Disruption of the 7SK snRNP complex has been associated with reversing HIV-1 latency (Stoszko et al, 2020), suggesting that targeting LARP7 LLPS could be a promising approach to developing novel anti-HIV-1 agents aimed at reactivating latent HIV-1 and hence contributing to eradication strategies.

LARP7, a member of the La and La-related protein (LARP) superfamily, plays a role in various RNA metabolic processes (Wang et al, 2020). LARP7 has been shown to promote the processing of U6 snRNA, essential for pre-mRNA splicing fidelity and male germ cell development, with dysregulation of this function linked to Alazami syndrome (Hasler et al, 2020). Recent studies have also implicated LARP7 in aging, heart failure, and other disorders (Yan et al, 2021; Yu et al, 2021). Our study highlights LLPS as an intrinsic characteristic of LARP7, potentially contributing to its diverse functions. These findings could offer new insights into the pathophysiology of LARP7-related diseases and suggest novel therapeutic targets.

In summary, our study unveils a novel mechanism by which the host proteins combat HIV-1 through phase separation. The discovery of phase separation involving LARP7 and Tat offers new perspectives for the development of therapeutic strategies that target HIV-1 infection and other LARP7-related diseases.

## Methods

**Reagents and tools table**

| Reagent/resource | Reference or source | Identifier or catalog number |
|---|---|---|
| **Experimental models** | | |
| HEK293T cells (*H. sapiens*) | ATCC | Cat# CRL-11268 |
| Tzm-bl cells (*H. sapiens*) | Prof. Lin Li (Beijing Institute of Microbiology and Epidemiology) | N/A |
| Human PBMCs | Schbio | PBMNC050C |
| Jurkat cells (*H. sapiens*) | Cell Resource Center, Peking Union Medical College | CloneE6-1 |
| SF9 | Cell Resource Center, Peking Union Medical College | 1101INS-PUMC000117 |

| Reagent/resource | Reference or source | Identifier or catalog number |
|---|---|---|
| **Recombinant DNA** | | |
| PCDH-CMV-3×Flag-LARP7(WT/Mut) | This paper | N/A |
| PCDH-CMV-3×Flag | This paper | N/A |
| pmCherry-C1-LARP7(WT/Mut) | This paper | N/A |
| pmEGFP-C1-Tat | This paper | N/A |
| PCDH-CMV-Cry2-mCherry | This paper | N/A |
| PCDH-CMV-Cry2-mCherry-LARP7(WT/Mut) | This paper | N/A |
| **Antibodies** | | |
| Mouse anti-GFP | Santa Cruz | Cat# sc-101525 |
| Rabbit anti-LARP7 | Proteintech | Cat# 17067-1-AP |
| Mouse anti-HIV1-Tat | Santa Cruz | Cat# sc-65912 |
| Mouse anti Cyclin T1 | Santa Cruz | Cat# sc-271348 |
| Mouse anti-Flag | Sigma-Aldrich | Cat# F3165 |
| Rabbit anti-CDK9 | Santa Cruz | Cat# sc-13130 |
| Goat anti-MEPCE | Novus | Cat# NB100-93420 |
| Mouse anti-NPM1 | Abcam | Cat# ab10530 |
| Goat anti-HEXIM1 | Abcam | Cat# ab58573 |
| Mouse anti-α-tubulin | Sigma-Aldrich | Cat# T5168 |
| FITC-CD4 antibodies | BioLegend | Cat# 357406 |
| **Oligonucleotides and other sequence-based reagents** | | |
| siRNA: human LARP7 (5′-ACAAGCGAGTAAACATATA-3′) | RiboBio | N/A |
| **RT-qPCR primers:** | | |
| Human LARP7 | F: 5′-CGGTCACGAGTTAAACAGGTG-3′ R: 5′-GCCTTCCAAATCAAGCTCTACAA-3′ | |
| Human GAPDH | F: 5′-GGAGCGAGATCCCTCCAAAAT-3′ R: 5′-GGCTGTTGTCATACTTCTCATGG-3′ | |
| HIV-1 *ltr* | F: 5′-TCTCTCGACGCAGGACTCGGC-3′ R: 5′-CCTTCTAGCCTCCGCTAGTCAAA-3′ | |
| HIV-1 *pol* | F: 5′-GCACTTTAAATTTTCCCATTAGTCCTA-3′ R: 5′-CAAATTTCTACTAATGCTTTTATTTTTC-3′ | |
| **Chemicals, enzymes, and other reagents** | | |
| BeyoGold His-tag Purification Resin | Beyotime Biotechnology | Cat# P2218 |
| Poly-D-lysine | Beyotime Biotechnology | Cat# ST508 |
| 1,6-Hexanediol | Sigma-Aldrich | Cat# 240117 |
| 2,5-Hexanediol | Sigma-Aldrich | Cat# H11904 |
| PEG-3000 | Sigma-Aldrich | Cat# 819015 |
| PIPES, free acid | Sango Biotech | Cat# A600719 |
| Recombinant human interleukin 2 | PEPROTECH | Cat# 200-02-50ug |
| Phytohemagglutinin (PHA) | Thermo Fisher Scientific | Cat#10576015 |

| Reagent/resource | Reference or source | Identifier or catalog number |
|---|---|---|
| PowerUp SYBR Green Master Mix | Thermo Fisher Scientific | Cat# A25742 |
| PrimeScript RT Reagent Kit | Takara | Cat# RR037A |
| Anti-Flag beads | Selleck | Cat# B23102 |
| Human CD4+ T Cell Isolation Kit | Miltenyi | Cat# 130-096-533 |
| Dual-Luciferase® Reporter Assay System | Promega | Cat# E1910 |
| **Bacterial and virus strains** | | |
| HIV R7-ΔEnv-GFP | Prof. Thomas J. Hope (Department of Cell and Molecular Biology, Feinberg School of Medicine, Northwestern University,) | N/A |
| VSV-GFP | Brain Case | Cat# BC-VSV |
| HSV-GFP | Brain Case | Cat# BC-HSV-Hs522 |
| **Software** | | |
| ImageJ | NIH | https://imagej.nih.gov/ij/ |
| GraphPad 8.0 | GraphPad Software | https://www.graphpad.com/ |
| FlowJo (v.10) | Flow Jo | https://www.flowjo.com/ |
| Volocity (×64) | Volocity Software | https://www.volocity4d.com/ |
| Pondr | Prediction of Intrinsically Unstructured Proteins | http://pondr.com/ |
| IUPred2A | Prediction of Intrinsically Unstructured Proteins | https://iupred2a.elte.hu/ |
| MEGA 7.0 | MEGA Software | https://www.megasoftware.net/ |

## Cell culture and transfection

Jurkat cells were cultured in RPMI 1640 medium (Invitrogen) supplemented with 10% fetal bovine serum (FBS), penicillin (100 U/mL), and streptomycin (100 mg/mL) at 37 °C, under 5% $CO_2$. HEK293T and Tzm-bl cells were cultured in Dulbecco's modified Eagle's medium (Invitrogen) containing 10% FBS and 1% penicillin–streptomycin.

Primary CD4+ T cells were isolated from healthy human PBMCs (purchased from SCHBIO) by using a CD4+ T Cell Isolation Kit (130-096-533, Miltenyi) according to the manufacturer's instructions. hPBMCs and primary CD4+ T cells were cultured in RPMI 1640 medium supplemented with 10% FBS, penicillin (100 U/mL), and streptomycin (100 mg/mL). Primary CD4+ T cells were supplied with 1/1000–1/2000 recombinant human interleukin 2 (200-02-50ug, PEPROTECH) to maintain proliferation viability,

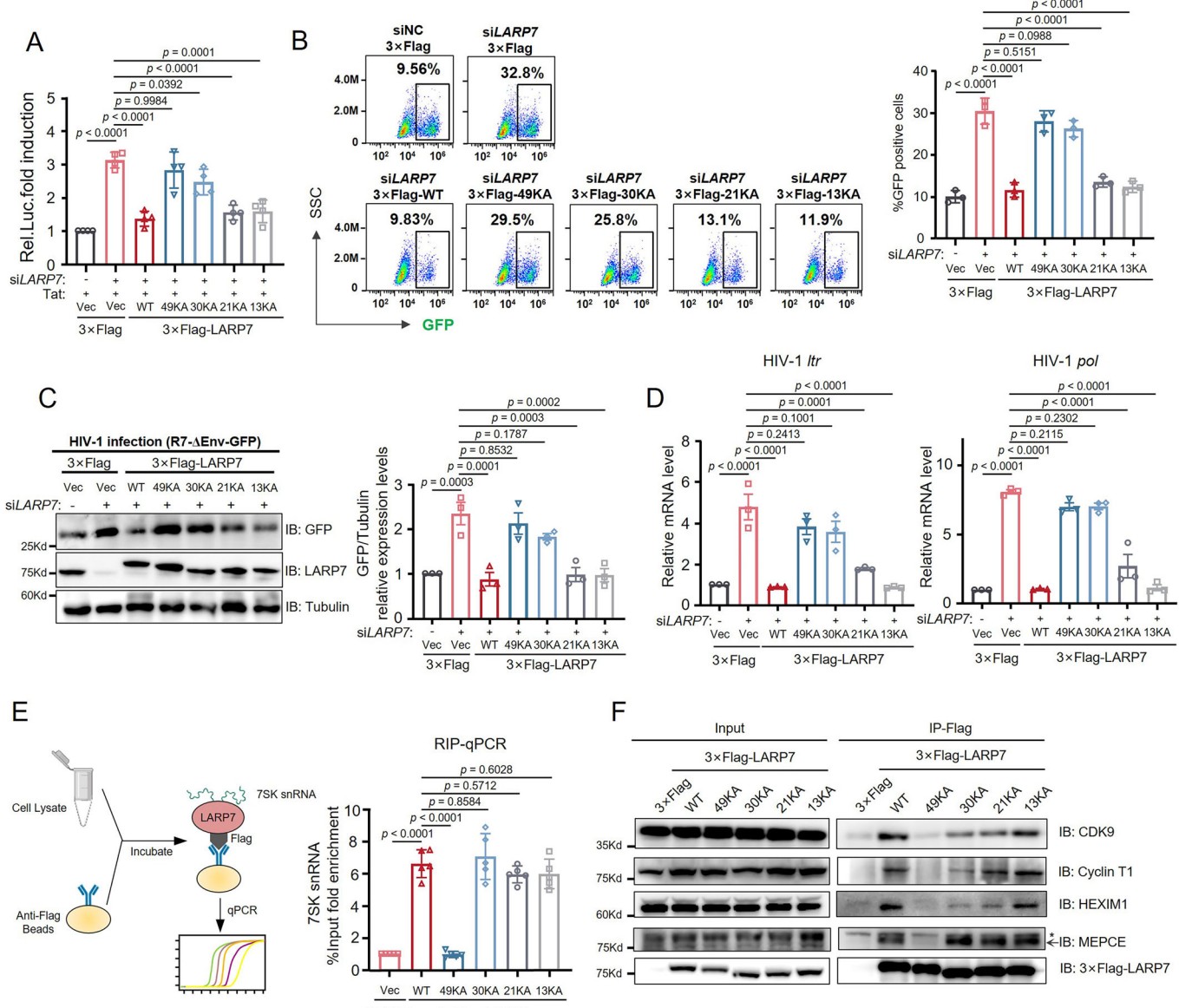

**Figure 4. LARP7 inhibits Tat-mediated HIV-1 infection through phase separation.**

(**A**) Dual-luciferase reporter assay performed in Tzm-bl cells stably expressing 3×Flag-LARP7(WT/mut). After transient knockdown of LARP7 for 24 h, mEGFP-Tat-expressing plasmid and pRL-TK (the *Renilla* luciferase reporter plasmid) were co-transfected into cells, and the luciferase activity was determined 24 h later. Each data point represents an independent biological replicate ($n = 4$). One-way analysis of variance (ANOVA) was used for statistical analysis, and exact $p$ values are represented in the figure, mean ± SEM. (**B**) LARP7 was transiently knocked down in Tzm-bl cells, stably expressing different LARP7 variants; then, cells were infected with HIV-1 R7-ΔEnv-GFP for 48 h. The percentage of GFP-positive cells were determined by FACS. Each data point represents an independent biological replicate ($n = 3$). One-way analysis of variance (ANOVA) was used for statistical analysis, and exact $p$ values are represented in the figure, mean ± SEM. (**C**) Western blotting analysis of the expression level of GFP in the cells in Fig. 4B (left). Quantification of the relative expression level of GFP and tubulin in the cells (right). The western blotting results were analyzed by using ImageJ software. Each data point represents an independent biological replicate ($n = 3$). One-way analysis of variance (ANOVA) was used for statistical analysis, and exact $p$ values are represented in the figure, mean ± SEM. (**D**) RT-quantitative PCR (qPCR) analysis of the mRNA levels of HIV-1 *ltr* and *pol* in the cells in Fig. 4B. Each data point represents an independent biological replicate ($n = 3$). One-way analysis of variance (ANOVA) was used for statistical analysis, and exact $p$ values are represented in the figure, mean ± SEM. (**E**) RNA immunoprecipitation-qPCR analysis of the ability of LARP7 (WT/mut) to bind 7SK small nuclear RNA. Each data point represents an independent biological replicate ($n = 5$). One-way analysis of variance (ANOVA) was used for statistical analysis, and exact $p$ values are represented in the figure, mean ± SEM. (**F**) Immunoprecipitation analysis of the interaction between LARP7 (WT/mut) and P-TEFb and other components of 7SK snRNP, *means the non-specific bands. Source data are available online for this figure.

and cultured with 1% Phytohemagglutinin (PHA) (#10576015, Thermo Fisher Scientific) for 3 days before HIV-1 infection.

HEK293T and Tzm-bl cells were transfected with plasmids using Lipofectamine 2000 (#11668019, Invitrogen) according to the manufacturer's instructions, and then cultured for 24–48 h to perform live-cell imaging or western blotting. All plasmids used were verified by sequencing.

## Viruses and infection

To establish stable virus-expressing cell lines, 10-cm plates of HEK293T cells were co-transfected with PCDH-CMV-3×Flag-Puro or PCDH-CMV-3×Flag-LARP7(Wt/Mut)- Puro, VSV-g, and psPAX2 by calcium phosphate. Viruses were harvested at 48 and 72 h post-transfection and precipitated by the addition of 5× PEG. Tzm-bl cells were infected by these retroviruses and screened with 2 μg/mL puromycin.

To produce a virus for infection, HIV-1 R7-ΔEnv-GFP was co-transfected with VSV-g into HEK293T cells. The viruses were then harvested by the method described above.

## Immunofluorescence

Jurkat cells were collected by centrifugation and adhered to poly-l-lysine-coated glass slides. Tzm-bl cells were treated with CSK buffer (10 mM PIPES, pH 7.0, 100 mM NaCl, 300 mM sucrose, 3 mM MgCl$_2$, 0.5% Triton X-100) for 5 min as described previously (Egloff et al, 2017). Cells were fixed with 4% PFA at room temperature for 15 min, washed three times with phosphate-buffered saline (PBS), and permeabilized with 0.5% Triton X-100 in PBS on ice for 10 min. After that, the cells were blocked in PBS containing 5% bovine serum albumin and 0.3% Triton X-100 for 1 h at room temperature. Next, cells were incubated antibodies in a blocking buffer at 4 °C, overnight. Following three washes with PBS, cells were incubated with fluorescent secondary antibodies in a blocking buffer for 1 h. Next, after three washes in PBS, cells were stained with DAPI dye for 15 min at room temperature. Then, the slides were mounted with a mounting agent (#ZLI-9556, ZSGB-BIO) and stored at 4 °C. Images were acquired by using a Delta Vision Microscope (GE Healthcare). Quantification of images was performed by using ImageJ software.

To quantify the number and fluorescence intensity of the puncta in cells, images were standardized with the same "Brightness/Contrast" range in ImageJ software and converted to an 8-bit format, threshold for puncta identification, and measured by the Analyze Particles function.

## Live-cell imaging

Cells were cultured in eight-well chambers and transfected with the indicated plasmids. After 24 h, the cells were examined using a 63× oil objective lens and a Delta Vision Microscope to capture timelapse images. The cells continued to be incubated at 37 °C under 5% CO$_2$.

## Photoinduced phase separation research system

The fusion proteins Cry2-mCherry and Cry2-mCherry-LARP7 were constructed using the PCDH-CMV-Puro plasmid. After plasmid transfection into Tzm-bl cells, cells were observed under the Delta Vision Microscope 24–48 h later. Throughout the imaging process, 488-nm light was used to stimulate droplet formation, and pictures were taken at the indicated intervals.

## Protein purification

The cDNAs encoding LARP7 and HIV-1 Tat were cloned into a modified version of the pFastBac HTB expression vector (10584-027; Invitrogen). The base vector was engineered to include either mCherry or mEGFP. The constructed vectors were transformed into Escherichia coli DH10Bac cells (#BC112-01, Biomed) for screening and blue/white colony selection for PCR verification. Positive clones were cultured overnight in liquid Luria-Bertani medium (containing 50 μg/ml kanamycin, 7 μg/ml gentamicin, and 10 μg/ml tetracycline) at 37 °C. Next, bacmids were transfected into SF9 cells using Cellfectin II Reagent (#10362100, Thermo Fisher Scientific) to produce a recombinant baculovirus that expressed the target gene. The transfected SF9 cells were incubated at 28 °C; the supernatant was collected 72 h post-transfection, centrifuged at 500×g for 5 min, and stored in the dark at 4 °C as P1 virus. Viral amplification was performed by infecting SF9 cells with P1 virus for 72 h, which was repeated to obtain P3 virus. Cells were harvested for protein purification after 72 h of P3 virus infection. The SF9 cells were centrifuged at 1000×g for 10 min and resuspended in His-binding buffer (50 mM Tris-HCl [pH 7.5], 500 mM NaCl, 5% glycerol [vol/vol], 10 mM imidazole, 1 mM phenylmethylsulfonyl fluoride [PMSF]), and placed on ice for 30 min, followed by sonication. Following centrifugation for 30 min at 12,000×g, the lysate was co-incubated with pre-equilibrated Ni-NTA agarose (G106-100; Genstar) at 4 °C for 2–3 h. Next, the lysate was added to a protein purification gravity column, and washed with buffer (50 mM Tris-HCl [pH 7.5], 500 mM NaCl, 5% glycerol [vol/vol], 30 mM imidazole, 1 mM PMSF). The protein was then eluted with buffer containing 50 mM Tris-HCl (pH 7.5), 500 mM NaCl, 5% glycerol [vol/vol], 500 mM imidazole, and 1 mM PMSF. The protein was concentrated using a 50-kD Amicon Ultra-15 device (UFC903096; Millipore; centrifugation for 5 min at 25,000×g). Subsequently, the supernatant was further purified using the GE AKTA fast protein liquid chromatography system. The collected fractions were concentrated and stored at −80 °C.

## In vitro droplet assays

Indicated concentrations of recombinant proteins were diluted in phase separation buffer containing 50 mM Tris-HCl (pH 7.5), 150 mM NaCl, and 1 mM dithiothreitol with 5% PEG-3000 as a crowding agent. The protein solution was immediately loaded into a glass-bottomed cell culture dish (NEST, 801002) and imaged by using the Delta Vision Microscope.

To determine partition coefficients, at least three images were analyzed, mean ± SEM is reported. Partition coefficients were calculated as the total fluorescence intensity of droplets divided by the bulk fluorescence intensity of the background.

## Fluorescence recovery after photobleaching

Tzm-bl or HEK293T cells were subjected to FRAP experiments 24–48 h post-transfection. FRAP was monitored in part of a cell containing condensates using a 488-nm laser at 100% power until the fluorescence intensities were 30% of the prebleached values, and

images were acquired every 2 s using a ZEISS confocal microscope (LSM 880). The data were analyzed by using ImageJ software and the fluorescence intensities were normalized to the prebleached values.

## RNA interference

Tzm-bl cells were transfected with siRNA using RNAiMax transfection reagent (#13778100, Thermo Fisher Scientific), and, 24 h later, transfected again with plasmids. The siRNA used was si-*LARP7* (5′-ACAAGCGAGTAAACATATA-3′).

## RNA extraction and quantitative PCR

Total RNAs were isolated from cells using TRI reagent. Total RNAs (500 ng) were reversed-transcribed to cDNA using the PrimeScript RT Reagent Kit (RR037A, Takara). mRNA expression was analyzed by qPCR with PowerUp SYBR Green Master Mix (A25742, Thermo Fisher Scientific) on an Applied Biosystems StepOnePlus system.

## Luciferase reporter assay

Tzm-bl cells were seeded in a 96-well plate and transfected with plasmids (pmEGFP-C1-*Tat* and pRL-*TK*) using Lipofectamine 2000. After 24 h, cells were harvested, and luciferase activities were measured using a dual-luciferase reporter assay system (#E1910, Promega) according to the manufacturer's instructions. All luciferase values were normalized by the activity of *Renilla* luciferase, and they are represented as the mean of a four set of experiments ± SEM.

## Flow cytometry analysis

CD4$^+$T cells were magnetically isolated from human PBMCs. In order to test the isolated efficiency, the CD4$^+$T cells were stained using FITC-CD4 antibodies (BioLegend, 357406, 1:200) at 4 ℃ in the dark. Then the cells were analyzed by flow cytometry.

Tzm-bl and Jurket cells infected by HIV-1 R7-ΔEnv-GFP virus were digested to achieve a single-cell suspension and washed twice with PBS. Then the cells were analyzed by flow cytometry.

All flow cytometry experiments were performed using the BD Accuri C6 Plus Flow Cytometer. The results were analysed using FlowJo (v.10).

## Immunoblotting and immunoprecipitation

Cells were lysed in 1 mL M2 lysate buffer (20 mM Tris-HCl, pH 7.5, 0.5% Nonidet P-40, 250 mM NaCl, 3 mM ethylenediaminetetraacetic acid, 3 mM ethylene glycol tetraacetic acid, and 2 mM dithiothreitol) supplemented with complete protease inhibitor cocktail for 30 min at 4 ℃, followed by sonication and centrifugation at 15,000×g for 20 min. Immunoprecipitation was performed with anti-FLAG M2 beads (#B23102; Selleck) for 8 h at 4 ℃. The immunoprecipitants were washed six times with M2 buffer and eluted with sodium dodecyl sulfate (SDS)-loading buffer by boiling for 20 min. The immunoprecipitated proteins were separated thereafter by SDS-polyacrylamide gel electrophoresis. Immunoblotting analysis was performed with primary antibodies and secondary antibodies conjugated to horseradish peroxidase.

## RNA immunoprecipitation-qPCR

HEK293T cells were transfected with plasmid [p3×Flag, p3×Flag-*LARP7*(WT/mut)], and cells were lysed 36 h later by the method described above with the addition of an RNase inhibitor. After combining with anti-FLAG M2 beads, the cell lysate was discarded, and TRI reagent was added to extract the RNA bound on the beads. The following steps were the same as described in section RNA extraction and quantitative PCR.

## Statistical analysis

All experiments were replicated independently at least three times. Statistical analysis was performed by using GraphPad Prism 8.0 software; statistical significance was calculated as indicated in the figure legends.

## Data availability

This study includes no data deposited in external repositories.

The source data of this paper are collected in the following database record: biostudies:S-SCDT-10_1038-S44319-025-00421-9.

## Peer review information

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

## Acknowledgements

This work was supported by grants from the National Science Fund for Excellent Young Scholars (81922051). We thank Liwen Bianji (Edanz) (www.liwenbianji.cn) for editing the language of a draft of this manuscript.

## Author contributions

**Zhuoxin Li:** Conceptualization; Data curation; Formal analysis; Validation; Investigation; Visualization; Methodology; Writing—original draft; Writing—review and editing; conceived the project, designed and performed the

experiments, analyzed the data and wrote the initial draft. **Xiya Fang**: Data curation; Formal analysis; Validation; Investigation; Visualization; Methodology; Writing—review and editing; performed the experiments, analyzed the data, and reviewed and proofread the manuscript. **Bing Zhao**: Data curation; Formal analysis; Investigation; Methodology; conceived the project, constructed the methods and performed the experiments. **Ran Liu**: Conceptualization; Data curation; Formal analysis; Investigation; Visualization; Methodology; Writing—review and editing; performed the qPCR experiments, analyzed the data, reviewed and proofread the manuscript. **Yezhuang Shen**: Data curation; Formal analysis; Investigation; Visualization; Methodology; Writing—review and editing; performed the qPCR experiments, analyzed the data, and reviewed and proofread the manuscript. **Tingting Li**: Conceptualization; Data curation; Formal analysis; Investigation; Methodology; conceived the project and performed the bioinformatic analyses. **Yining Wang**: Data curation; Validation; Investigation; Methodology; Writing—review and editing; reviewed and proofread the manuscript. **Zenglin Guo**: Data curation; Formal analysis; Validation; Investigation; Methodology; Writing—review and editing; performed the western blot experiments, reviewed and proofread the article. **Wen Wang**: Data curation; Formal analysis; Validation; Investigation; Methodology; Writing—review and editing; help perform the western blot experiments. **Biyu Zhang**: Data curation; Validation; Investigation; Visualization; Methodology; Writing—review and editing; help perform the western blot experiments. **Qiuying Han**: Data curation; Validation; Visualization; Methodology; help perform the experiments in cells. **Xin Xu**: Data curation; Validation; Visualization; Methodology; help collect and analyse the immunofluorescence sample data. **Kai Wang**: Data curation; Formal analysis; Validation; Visualization; Methodology; help collect and analyse the immunofluorescence sample data. **Libing Yin**: Data curation; Formal analysis; Validation; Methodology; Project administration; help collect and analyse the FACS data. **Weili Gong**: Data curation; Validation; Visualization; Project administration; managed and coordinated the planning and execution of this research. **Ailing Li**: Supervision; Validation; Visualization; Project administration; provided scientific advice for this study. **Tao Zhou**: Conceptualization; Formal analysis; Supervision; Validation; Visualization; Methodology; Project administration; Writing—review and editing; provided scientific advice for this study. **Teng Li**: Conceptualization; Formal analysis; Supervision; Validation; Visualization; Methodology; Writing—original draft; Project administration; Writing—review and editing; conceived the project, supervised the project and acquired funding for this study. **Weihua Li**: Conceptualization; Supervision; Validation; Investigation; Writing—original draft; Project administration; Writing—review and editing; supervised the project and revised the manuscript.

Source data underlying figure panels in this paper may have individual authorship assigned. Where available, figure panel/source data authorship is listed in the following database record: biostudies:S-SCDT-10_1038-S44319-025-00421-9.

## Disclosure and competing interests statement

The authors declare no competing interests.

# Expanded View Figures

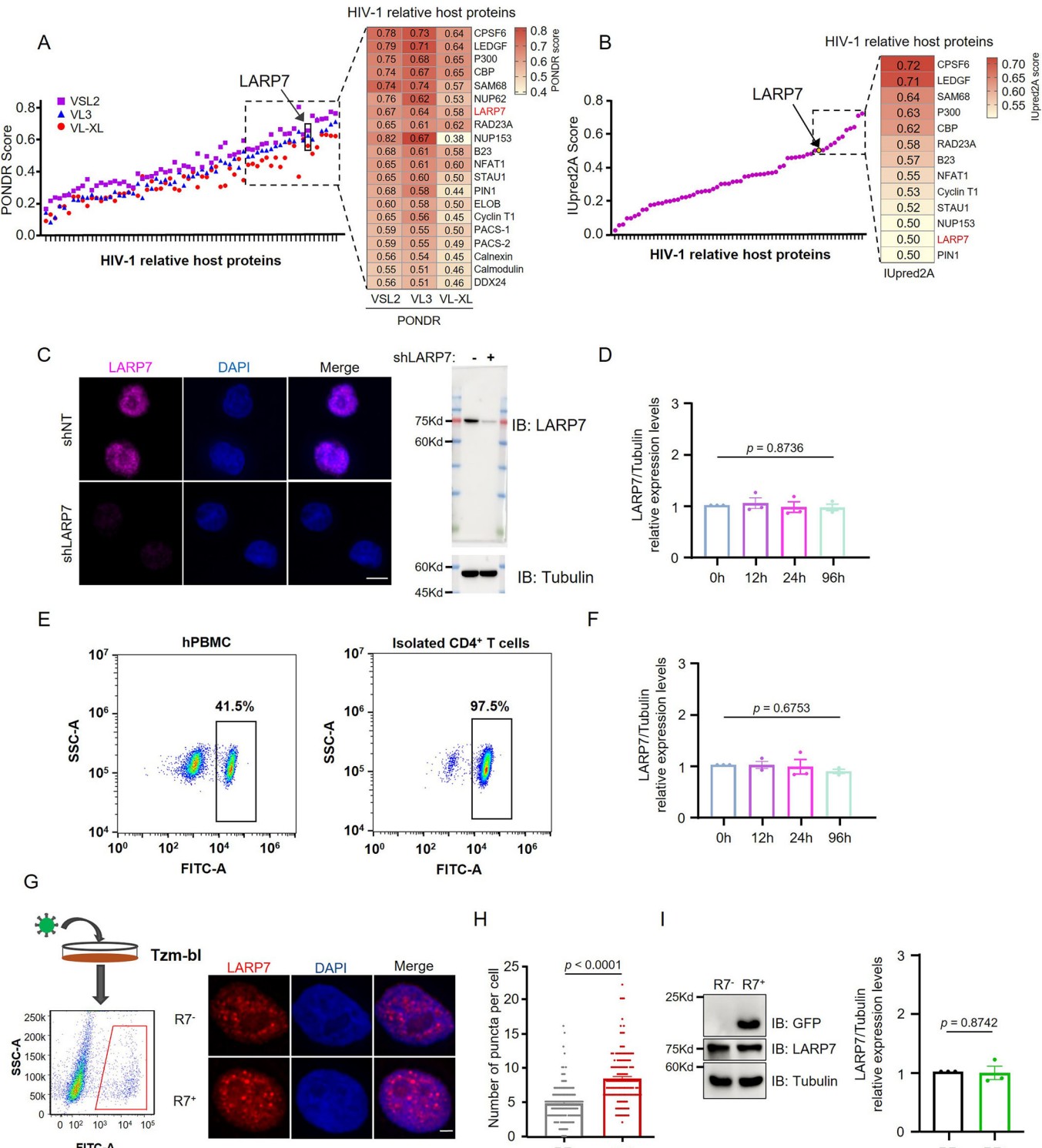

**Figure EV1.   LARP7 undergoes liquid-liquid phase separation after HIV-1 infection.**

(**A**) IDR content scoring and ranking of host proteins reported to be involved in HIV-1 replication using the predictor PONDR. (**B**) IDR content scoring and ranking of host proteins were reported to be involved in HIV-1 replication using the predictor IUPred2A. (**C**) Western blotting and immunofluorescence analyses (Scale bar, 5 μm) for Jurkat cells, testing the specificity of the anti-LARP7 antibody. (**D**) Quantification of the relative expression level of LARP7 and tubulin in Jurkat cells at different times after HIV-1 R7-ΔEnv-GFP infection. The western blotting results were analyzed by using ImageJ software. Each data point represents an independent biological replicate ($n = 3$). One-way analysis of variance (ANOVA) was used for statistical analysis, and exact $p$ values are represented in the figure, mean ± SEM. (**E**) FACS analysis of the percentage of primary CD4$^+$ T cells isolated from human primary peripheral blood mononuclear cells of healthy individuals. (**F**) Quantification of the relative expression level of LARP7 and tubulin in primary CD4$^+$ T cells at different times after HIV-1 R7-ΔEnv-GFP infection. The western blotting results were analyzed by using ImageJ software. Each data point represents an independent biological replicate ($n = 3$). One-way analysis of variance (ANOVA) was used for statistical analysis, and exact $p$ values are represented in the figure, mean ± SEM. (**G**) Left, flow cytometry-sorted GFP-positive and GFP-negative Tzm-bl cells 24 h after HIV-1 R7-ΔEnv-GFP infection. Right, immunofluorescence of LARP7 in sorted cells. Scale bar, 2 μm. (**H**) Quantification of the number of puncta in individual cells from Fig. EV1G; 150 cells were analyzed in each group. A two-tailed unpaired student's $t$-test was used for statistical analysis, and exact $p$ values are represented in the figure, mean ± SEM. (**I**) Left, western blotting analysis of LARP7 expression in sorted Tzm-bl cells from Fig. EV1G. Right, quantification of the relative expression level of LARP7 and tubulin. The western blotting results were analyzed by using ImageJ software. Each data point represents an independent biological replicate ($n = 3$). A two-tailed unpaired student's $t$-test was used for statistical analysis, and exact $p$ values are represented in the figure, mean ± SEM.

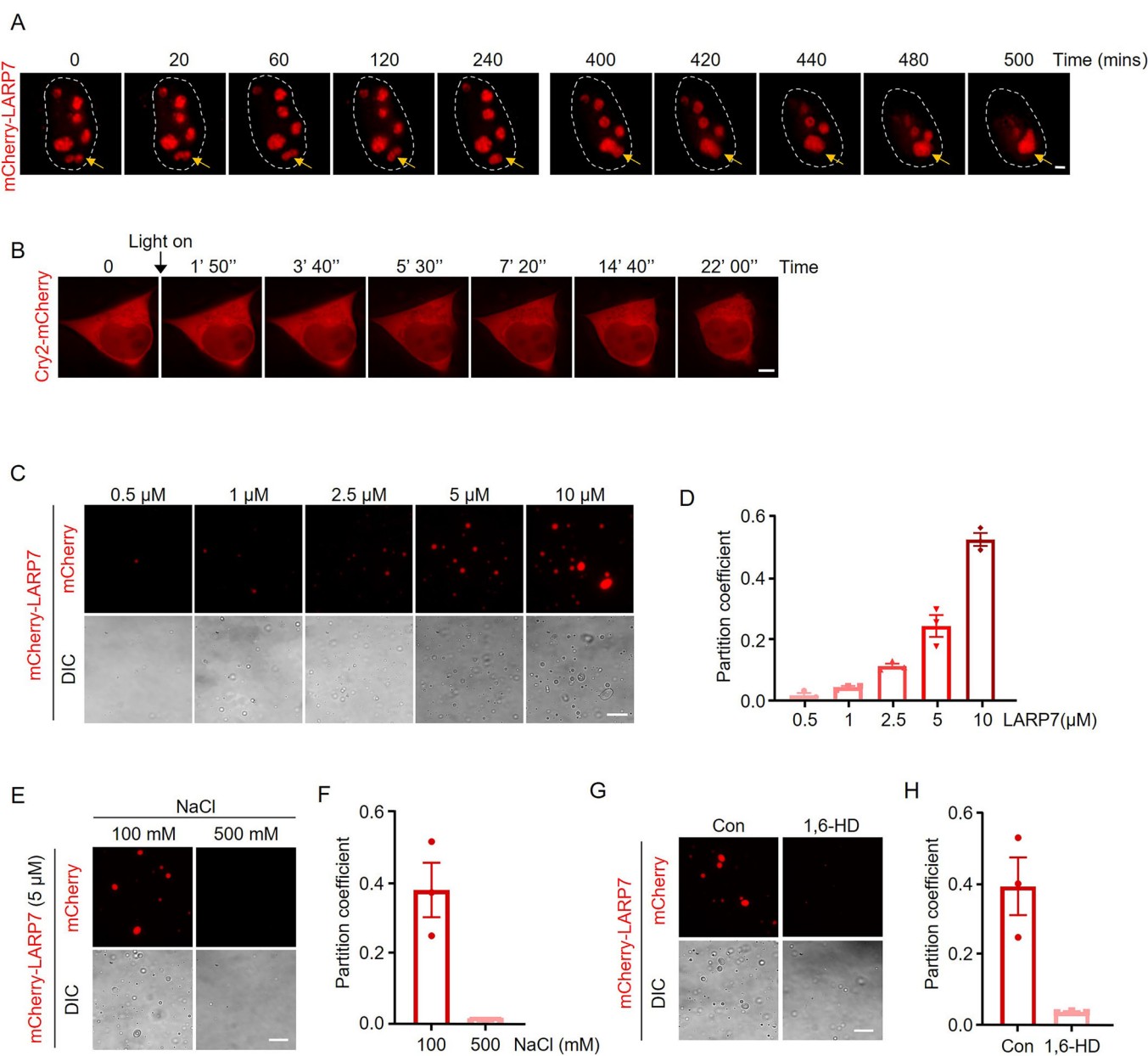

**Figure EV2. LARP7 undergoes liquid-liquid phase separation *in cellulo* and in vitro.**

(A) Timelapse imaging of cells expressing mCherry-LARP7, subjected to laser excitation every 20 min. A droplet fusion event occurs, indicated by the yellow arrows. Scale bar, 5 μm. (B) Timelapse imaging of the cell expressing Cry2-mCherry during stimulation by blue light. Scale bar, 5 μm. (C) Representative images of mCherry-LARP7 droplets with the indicated concentrations. Scale bar, 20 μm. (D) Quantification of partition coefficients for individual conditions in Fig. EV2C. Partition coefficients were calculated as the total fluorescence intensity of droplets divided by the bulk fluorescence intensity of the background. Three random fields were analyzed in each experiment, mean ± SEM. (E) Representative images of mCherry-LARP7 droplets (5 μM) with different added concentrations of NaCl. Scale bar, 20 μm. (F) Quantification of partition coefficients for individual combinations in Fig. EV2E. Three random fields were analyzed in each experiment, mean ± SEM. (G) Representative images of mCherry-LARP7 droplets (5 μM) with 6% 1,6-HD added. Scale bar, 20 μm. (H) Quantification of partition coefficients for the experiment in Fig. EV2G. Three random fields were analyzed in each experiment, mean ± SEM.

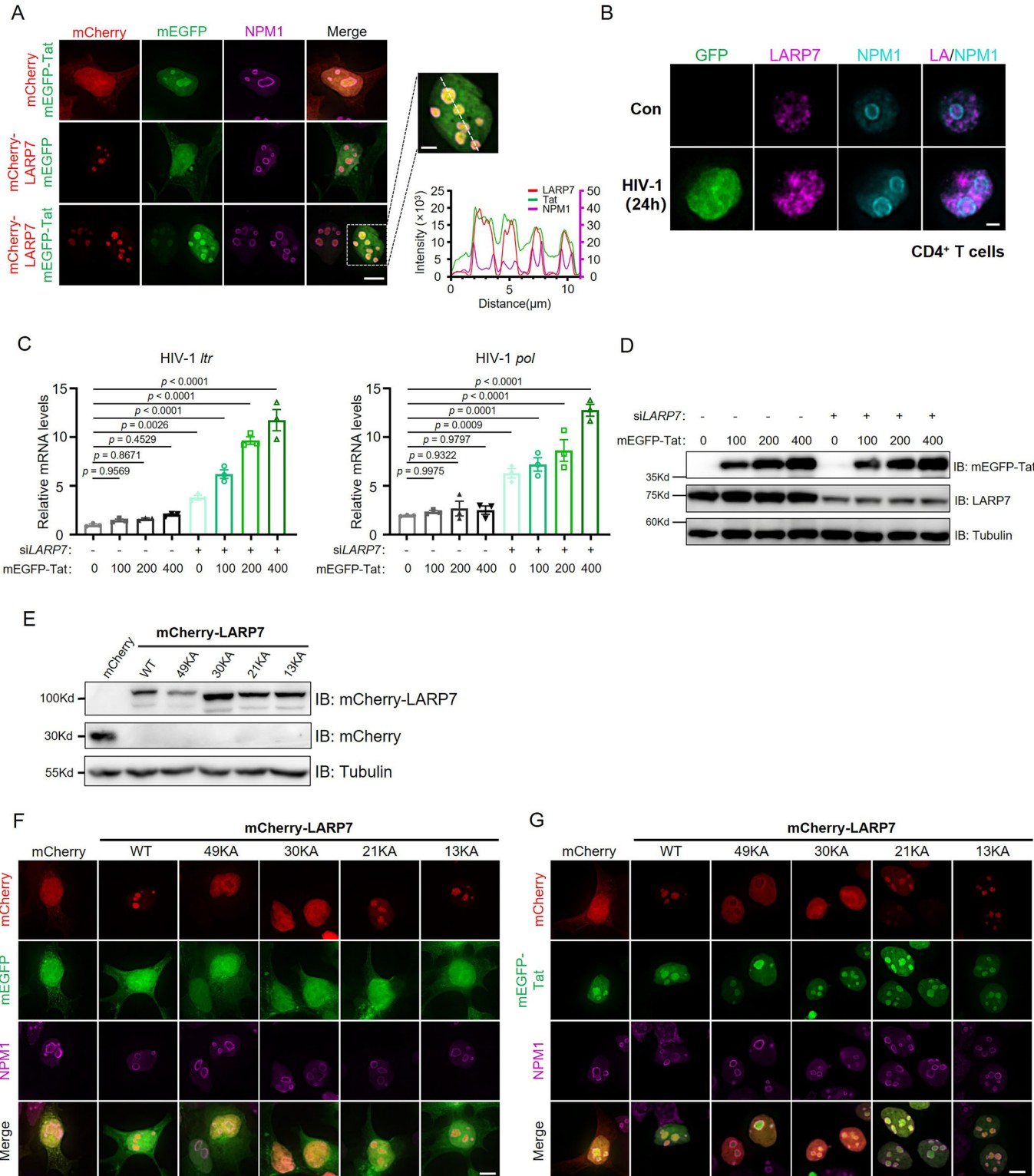

◀ **Figure EV3. LARP7 forms liquid-liquid phase separation droplets with Tat.**

(A) Left, representative images of HEK293T cells expressing mCherry-LARP7 and mEGFP-Tat, separately and together. The cells were stained with an anti-NPM1 antibody. Scale bar, 10 µm. Right, the co-localized droplets are enlarged, scale bar, 5 µm. Fluorescence intensity profiles are shown for mCherry-LARP7 (red curve), mEGFP-Tat (green curve) and NPM1 (magenta curve) on the dotted line indicated in the image. (B) Immunofluorescence of LARP7 and NPM1 in primary CD4$^+$ T cells 24 h after infection by HIV-1 R7-ΔEnv-GFP. Scale bar, 2 µm. (C) RT-qPCR analysis of the mRNA levels of HIV-1 *pol* and *ltr* in individual cells. Each data point represents an independent biological replicate ($n = 3$). One-way analysis of variance (ANOVA) was used for statistical analysis, and exact *p* values are represented in the figure, mean ± SEM. (D) Western blotting analysis of the effect of LARP7 knockdown and the expression of mEGFP-Tat in individual cells in Fig. EV3C. (E) Western blotting analysis of mCherry-tagged LARP7 variants expressed in HEK293T cells. (F) Representative images of HEK293T cells transfected with mEGFP and mCherry-LARP7 (WT or IDR mutants). Scale bar, 10 µm. (G) Representative images of HEK293T cells transfected with mEGFP-Tat and mCherry-LARP7 (WT or IDR mutants). Scale bar, 10 µm.

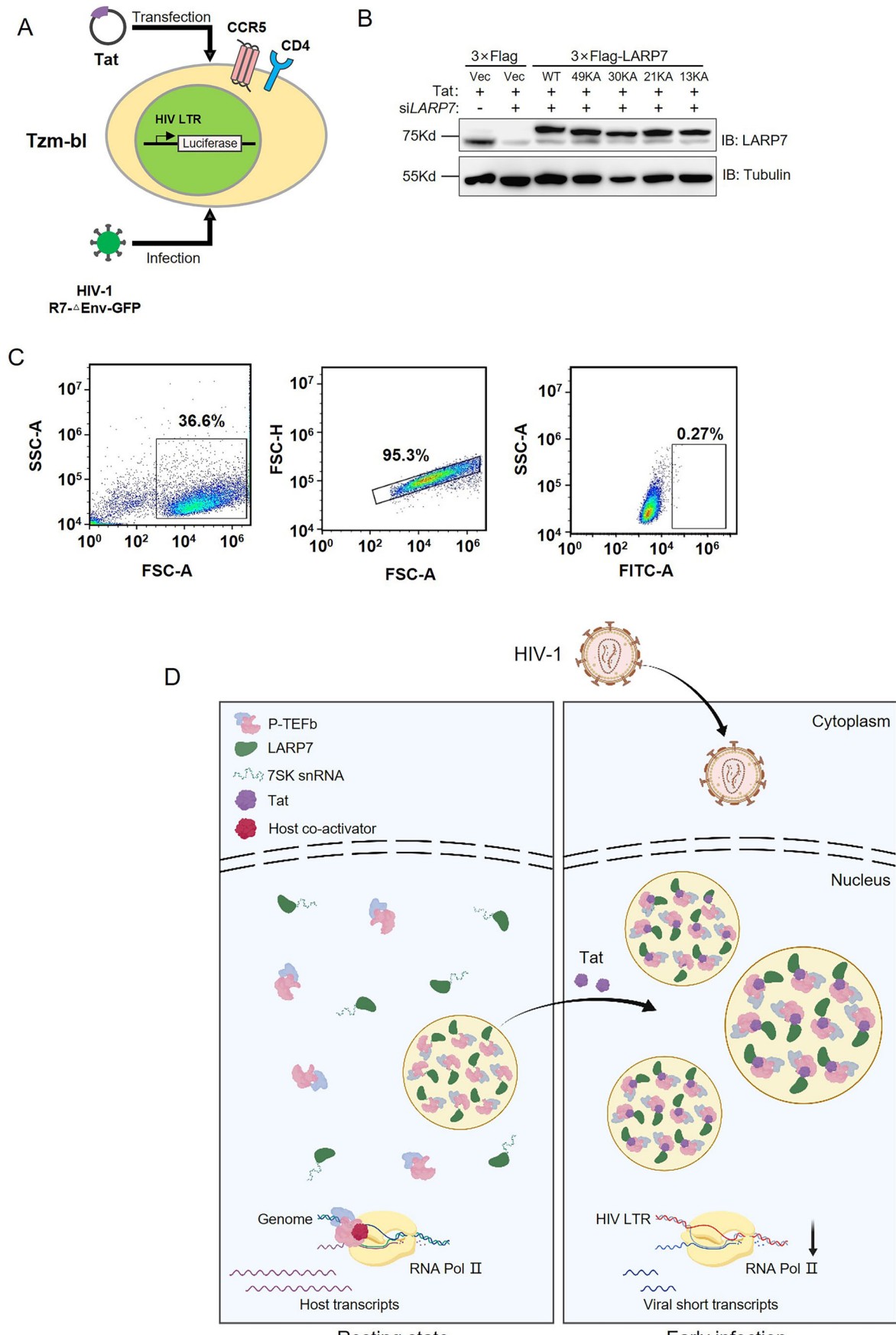

**Figure EV4. LARP7 inhibits Tat-mediated HIV-1 infection through phase separation.**

(A) Schematic illustrating the dual-luciferase reporter assay and HIV-1 infection experiments in the Tzm-bl cell line. (B) Western blotting analysis of the effect of LARP7 knockdown and the expression of LARP7 variants in individual cells in Fig. 4A. (C) LARP7 was transient knockdown in Tzm-bl cells stably expressing different LARP7 mutants, infected the cells with HIV-1 R7-ΔEnv-GFP, and detected GFP expression 48 h after infection using FACS. (D) Model in which LARP7 forms condensates with the 7SK snRNP complex as well as Tat, confining Tat to inhibit HIV-1 transcriptional activation.

