## [Peer Review File · EMBO Reports]

Liquid-liquid phase separation of LARP7 restrains HIV-1 replication.

Zhuoxin Li, Xiya Fang, Bing Zhao, Ran Liu, Yezhuang Shen, Tingting Li, Yining Wang, Zeng-Lin Guo, Wen Wang, Biyu Zhang, Qiuying Han, Xin Xu, Kai Wang, Li Yin, Wei Gong, Ai-Ling Li, Tao Zhou, Teng Li, and Weihua Li

Corresponding author(s): Weihua Li (whli@ncba.ac.cn) , Teng Li (tnli@ncba.ac.cn)

Review Timeline:

Submission Date:	1st May 24
Editorial Decision:	4th Jun 24
Revision Received:	3rd Dec 24
Editorial Decision:	12th Feb 25
Revision Received:	25th Feb 25
Accepted:	5th Mar 25

Editor: Achim Breiling

Transaction Report:

Dear Dr. Li,

Thank you for the transfer of your manuscript to EMBO reports. I have now received the reports from the three referees that were asked to evaluate your study, which can be found at the end of this email. As you will see, the referees have several comments, concerns, and suggestions, indicating that a major revision of the manuscript is necessary to allow publication of the study in EMBO reports. As the reports are below, and all the concerns need to be addressed, I will not detail them here.

Given the constructive referee comments, I would like to invite you to revise your manuscript with the understanding that the concerns of the referees must be addressed in the revised manuscript and in a detailed point-by-point response. Acceptance of your manuscript will depend on a positive outcome of a second round of review. It is EMBO reports policy to allow a single round of revision only and acceptance of the manuscript will therefore depend on the completeness of your responses included in the next, final version of the manuscript.

Moreover, please have your revised manuscript carefully edited by a native speaker before re-submission.

- 1) a .docx formatted version of the final manuscript text (including legends for main figures, EV figures and tables), but without the figures included. Figure legends should be compiled at the end of the manuscript text.
- 2) individual production quality figure files as .eps, .tif, .jpg (one file per figure), of main figures and EV figures. Please upload these as separate, individual files upon re-submission.

- 4) a complete author checklist, which you can download from our author guidelines (<https://www.embopress.org/page/journal/14693178/authorguide>). Please insert page numbers in the checklist to indicate where the requested information can be found in the manuscript. The completed author checklist will also be part of the RPF.

- 5) that primary datasets produced in this study (e.g. RNA-seq, ChIP-seq, structural and array data) are deposited in an

appropriate public database. If no primary datasets have been deposited, please also state this in a dedicated section (e.g. 'No primary datasets have been generated and deposited'), see below.

The accession numbers and database should be listed in a formal "Data Availability" section (placed after Materials & Methods) that follows the model below. This is now mandatory (like the COI statement). Please note that the Data Availability Section is restricted to new primary data that are part of this study. This section is mandatory. As indicated above, if no primary datasets have been deposited, please state this in this section

Data availability

8) Regarding data quantification and statistics, please make sure that the number "n" for how many independent experiments were performed, their nature (biological versus technical replicates), the bars and error bars (e.g. SEM, SD) and the test used to calculate p-values is indicated in the respective figure legends (also for EV figures and all those in an Appendix). Please also check that all the p-values are explained in the legend, and that these fit to those shown in the figure. Please provide statistical testing where applicable. Please avoid the phrase 'independent experiment', but clearly state if these were biological or technical replicates. Please also indicate (e.g. with n.s.) if testing was performed, but the differences are not significant. In case n=2, please show the data as separate datapoints without error bars and statistics. See also: <http://www.embopress.org/page/journal/14693178/authorguide#statisticalanalysis>

9) Please add scale bars of similar style and thickness to microscopic images, using clearly visible black or white bars (depending on the background). Please place these in the lower right corner of the images themselves. Please do not write on or near the bars in the image but define the size in the respective figure legend.

10) Please also note our reference format:

12) We now use CRedit to specify the contributions of each author in the journal submission system. CRedit replaces the author contribution section. Please use the free text box to provide more detailed descriptions and do not provide your final manuscript text file with an author contributions section. See also our guide to authors: <https://www.embopress.org/page/journal/14693178/authorguide#authorshipguidelines>

13) We would encourage you to use 'Structured Methods', our new Materials and Methods format. According to this format, the

Materials and Methods section should include a Reagents and Tools Table (listing key reagents, experimental models, software, and relevant equipment and including their sources and relevant identifiers), uploaded as separate file, followed by a Methods and Protocols section in which we encourage the authors to describe their methods using a step-by-step protocol format with bullet points, to facilitate the adoption of the methodologies across labs. More information on how to adhere to this format as well as downloadable templates (.doc or .xls) for the Reagents and Tools Table can be found in our author guidelines (section 'Structured Methods'):

14) Please order the manuscript sections like this, using these names:

Title page - Abstract - Keywords - Introduction - Results - Discussion - Methods - Data availability section - Acknowledgements - Disclosure and Competing Interests Statement - References - Figure legends - Expanded View Figure legends

I look forward to seeing a revised version of your manuscript when it is ready. Please let me know if you have questions or comments regarding the revision.

Yours sincerely,

Referee #1:

Li et al. propose that HIV-1 infection induces liquid-liquid phase separation (LLPS) of the host protein LARP7, forming condensates that sequester P-TEFb and the viral Tat protein, thereby inhibiting viral transcription. They argue that LARP7 and Tat co-condense in vitro and in cellulo (the term "in vivo" is incorrect and should be avoided), with LARP7 intrinsically disordered region (IDR) containing 49 lysines being essential for this phase separation. The authors suggest that targeting the phase separation properties of LARP7 could offer new anti-HIV-1 therapeutic strategies. However, they do not investigate how viral infection is subsequently established (LARP7/Tat ratio) and, more interestingly, how this proposed mechanism would affect viral transcription during latency reactivation.

The work would benefit from studies in primary cells and with wild-type virus infection.

The text is poorly edited, and figures (especially EVs) are often not fully commented in the results section. Controls are frequently missing, and in general the work is not of sufficient quality to be published in its current state.

Major Points:

Figure 1

"After HIV-1 infection, there was a notable rise in the number of LARP7 puncta that had fluorescence intensity higher than those of their surroundings."

Since the authors do not show the mean fluorescence intensity of LARP7 puncta but only their number, this sentence is not correct and has to be modified. Also, it is not clear how was this comparison, reported in Fig1c performed? How do the authors know which cells are HIV-1 infected? In the population of cells that they infected and assayed, not all cells (presumably) have HIV-1. It is unclear if the HIV-1 infected cells were compared to surrounding cells?

In Figure 1D at 96 hours, when infection peaks, LARP7 levels seem lower compared to the 0-hour time point. Could the authors provide densitometry of the WB replicates?

It would be informative to see the same analysis (WB and quantification) performed in Jurkat cells and compared to IF with LARP7 staining in GFP- and GFP+ cells).

At what time point of infection are 1,6-hexanediol (HD) treatments added?

Clarification needed for the correct interpretation of this OPTOIDR system. It seems that the authors somehow do not fully understand the principle. : "To further investigate the LLPS features of LARP7, we adopted a photo-induced phase separation research system, in which a photo-sensitive protein, Cry2, was fused with mCherry-LARP7. When blue light is on, mCherry-LARP7 molecules are brought closer to each other (Figure 1G)(Shin, Berry et al., 2017)."

The sentence "And Cry2-mCherry could not form droplets after the blue light stimulation." should be corrected. It starts with "And" and contains a typo ("after the after"). Proper text editing is necessary.

The authors use inconsistent concentrations of 1,6-HD.

Concluding sentence: "Taken together, the results above suggest that LARP7 undergoes LLPS both in vivo and in vitro, and the phase separation of LARP7 is induced when cells were infected by HIV-1." should be corrected to "in cellulo."

The term "partition coefficient (EV_J-N)" is mentioned without explanation in the text.

Figure 2

The first sentence is incomplete: "As a core component of the 7sk snRNP, LARP7 maintains the stability of the complex, which prevents P-TEFb from activating (Krueger et al., 2008)."

Figure 2B-2D: It is not clear if P-TEFb is entirely sequestered within LARP7 condensates, or if there is a separate pool of P-TEFb that interacts with Tat and not with LARP7. Therefore, a triple staining would be more appropriate to assess co-condensates of Tat, LARP7, and Cyclin T1.

Figure 2E: Condensates in HEK cells have a completely different morphology and are very large and few, unlike observations from HIV-1 relevant cells, Jurkat.

Figure 3

How were the defined lysine sets selected for the mutation experiments - it is not clear on what basis have the authors selected 30, 21 and 13 lysine residues (our of 49) for the downstream analysis.

Panel F: The stainings are really strange. To me those look like nucleoli rather than LLPS structures. The authors should use the B23 (nucleophosmin) or nucleolin to verify this. Moreover, these structures do not correlate to what is shown in Figure 2.

Figure 4

All immunoblots lack molecular weight markers.

Clarify Tat transfection versus cell infection and luciferase readings.

Figure 4B: The FLAG immunoblot is missing.

Figures 4B, C: What is the second band in the LARP7 WB?

Figure 4C: The FLAG immunoblot is missing.

Figure 4D: The LARP7 immunoblot is missing.

Reverse IPs are needed to confirm that IDR lysine residues are crucial for the interaction between LARP7 and components of P-TEFb.

Referee #2:

Li et al characterize how LARP7 undergoes LLPS in HIV infected cells and how this affects viral replication. They show that LARP7 puncta are induced upon HIV infection. Li et al then propose that these puncta are formed via LLPS because their abundance is decreased when cells are treated with 1,6-Hexanediol. They also show that the LARP7 puncta can fuse and have liquid-like properties in FRAP experiments. Cyclin T1 and HIV Tat were also in LARP7 puncta. In addition, Tat and LARP7 formed droplets in vitro. Therefore, the authors have shown that Tat, Cyclin T and LARP7 can form LLPS-like puncta.

A difficult challenge for many proposed LLPS systems is showing their functional relevance in a living cell. The authors have attempted to address this by mutating the large number of lysines in the LARP7 linker region between the La module and RRM2. They mutated 49, 30, 21 or 13 lysine residues to alanine and analyzed whether this affects LARP7 puncta formation or HIV transcription. The role for these lysine residues in LARP7 function is not clear, though many appear to evolutionarily conserved. The 49KA or 30KA mutants did not form puncta and did not inhibit HIV gene expression as efficiently as wild type LARP7. Proteins with 21 or 13 lysines mutated had activity more similar to the wild type protein. The 49KA and 30KA mutant proteins had reduced interaction with CDK9 and Cyclin T1, leading to the question of whether this is a specific effect on LLPS or whether the mutant proteins have lost several functions due to the mutations in the conserved linker region (see below).

Major comments:

1. The major point of interpretation is whether the lysines in the linker region have a function for LARP7 independent of LLPS and how this could affect its function in regulating transcription. The authors should show whether mutating these lysines have affected LARP7 binding to 7SK RNA. Previous reports have found that the basic residues in the central linker of LARP7 may contribute to 7SK binding (reviewed in RNA Biol. 2021;18:290-303), so these mutations may have a global effect on LARP7 activity that is independent from LLPS. The authors should also analyse whether these mutations have decreased interaction with MePCE. Based on the decreased interaction with CDK9 and Cyclin T1, it is possible that the lysine mutations have decreased the ability of LARP7 to interact with the 7SK complex due to decreased RNA binding. It is essential to rule out possibilities other than LLPS formation that the conserved linker region may contribute to.
2. Figure 4C needs to be quantified in multiple independent experiments. The magnitude of the effect of LARP7 on HIV transcription appears to be small (3-fold in Figure 4B, maybe less in Figure 4C). How important is this phenotype for HIV transcription? Measuring viral replication using full length HIV would be more sensitive than the GFP or luciferase reporter experiments.
3. The model proposed by the authors needs to be refined. Tat forms a positive regulatory circuit in which low levels of full length HIV transcription lead to a small pool of Tat which then promotes more efficient transcription and increased Tat levels and therefore high levels of processive HIV transcription. The authors propose at the end of the results section that early in HIV infection, low levels of Tat is trapped with P-TEFb in LARP7 droplets, which would make it inactive. This is hard to reconcile with the positive regulatory model in which low levels of Tat induce viral transcription. The role of Tat in regulating HIV transcription is

complex and there is no analysis of how it binds to HIV RNA or leads to Pol II phosphorylation in the experimental conditions in this manuscript. It is also not clear how HIV infection leads to the induction of LARP7 foci. Without more information, the model is currently incomplete and it is hard to determine the functional role of LARP7 puncta based only on the lysine mutations in LARP7.

Referee #3:

General comment:

In this manuscript, Li et al. explored the functional interaction between HIV-1 Tat protein and LARP7, a subunit of the 7SKsnRNP which sequesters and inhibits PTEFb kinase activity. They show that LARP7 has an intrinsic ability to undergo liquid-liquid phase separation (LLPS). They identified the intrinsically disordered region within and the associated lysine residue responsible for the formation of LARP7 droplets. LARP7 condensates are increased upon HIV-1 infection. The HIV-1 viral transactivator Tat and Cyclin T1, the regulatory subunit of PTEFb, are associated with LARP7 droplets. Finally, they show that LARP7-mediated inhibition of Tat activation of HIV-1 transcription is LARP7 IDR-dependent. The topic is of importance not only to the field of HIV but also in our understanding of PTEFb regulation. However, as it stands, the manuscript suffers from lack of important controls and additional experiments to support the authors' conclusions. Additionally, the manuscript needs language and grammar editing.

Major comments:

1*Figure 1 and EV1: the concern here is lack of controls both for the anti-LARP7 antibody and for the specificity towards HIV-1. Thus, an experiment showing the staining with anti-LARP7 in control cells versus cells transfected with LARP7 specific siRNA is needed. The use of VSV and HSV is highly appreciated. However, use of HIV-1 Tat minus virus or lentiviral vector and viral like particles (VLPs) will be of importance to assess what triggers the formation of LARP7 droplets. Indeed, enhanced LARP7 droplets are observed at 12hrs post infection where Tat has not yet been expressed. Interestingly, the viral protein R (VPR) has been shown to target CTIP2 which associates with and regulates 7SKsnRNP (Eilebrecht S. et al. NAR. 2014).

2*Figure 2E, 3F, EV2: The concern here is about the subcellular localization of ectopically expressed LARP7 and Tat. Indeed, both LARP7 and Tat are RNA binding proteins. Their overexpression results in accumulation in RNA enriched organelles such as the nucleolus. Thus, use of Tat and LARP7 mutants lacking RNA binding domains is required.

It has been shown that Tat displaces hexim1 from the 7SKsnRNP and forms a stable complex with 7SKRNA/LARP7/MEPCE through direct interaction with 7SKRNA (Sobhian et al. Mol. Cell. 2010; Muniz L. et al. Plos Pathog. 2010 and others). Thus, one question is the presence of HEXIM1 in LARP7 droplets containing Tat and CyclinT1. This should be addressed by assessing the presence of HEXIM1 in LARP7 droplets containing Tat and CyclinT1.

CyclinT1 has been shown to undergo LLPS (Lu H. et al. Nature 2018). It will be important to analyze whether TatC22G mutant, unable to bind CyclinT1, localizes to LARP7 droplets containing CyclinT1.

3*Figure 3: Results shown in 3F are of major concern (see comment 2). Indeed, LARP7 and Tat colocalize only when they both concentrate in the nucleolus. The use of LARP7 lacking its RNA binding domain is important. This mutant should not localize to the nucleolus but should still be able to form droplets. Question is whether this mutant is still able to trap Tat into the droplets.

4*Figure 4: In absence of the above proposed experiment, the functional results shown in figure 4 are hard to interpret. Indeed, as previously shown LARP7 knockdown will lead to 7SK RNA degradation and increased active PTEFb which explains the increased Tat transcriptional activity towards the viral LTR. An important experiment to be performed is to ask whether RNAPII associated with LARP7 droplets containing Tat is phosphorylated at Ser5 or Ser2. This can be performed with specific antibodies.

To exclude artificial aggregates one may consider using different, shorter tags or an otherwise different detection system (even simple tags such as HA) expressed from a weak to moderate promoter (not CMV or SV40 for instance but rather LTR driven or inducible but weakly to moderately induced).

Introduction

1. « Specifically speaking, when Tat functions normally, HIV-1 enters replication cycle, whereas when Tat dependent transcription is inhibited, HIV-1 primarily enters dormancy, which is the main obstacle for HIV-1 eradication (Razooky, Pai et al., 2015). »

The HIV promoter contains binding sites for host cell transcription factors such as NFκB and SP1, driving basal levels of transcription. Tat in this system is considered as a positive feedback loop. Latency may thus not be solely due to lack or presence of Tat activity.

Please consider the following paper: Jordan et al. The EMBO Journal 22(8):1868-77.

2. "Evidence showed that LARP7 also plays an inhibitive role in HIV-1 replication."

Please provide a reference

3. "Additionally, targeting the phase separation of LARP7 could potentially serve as a novel approach for treating AIDS."

It is unclear how LARP7 LLPS properties may be targeted for treating AIDS.

Methods

1. "And Tzm-bl cells were treated with CSK buffer (10mM PIPES PH7.0, 100mM NaCl, 300mM sucrose, 3mM MgCl₂, 0.5% Triton X-100) for 5 mins as described previously."

Please provide a reference.

Referee 1#

Li et al. propose that HIV-1 infection induces liquid-liquid phase separation (LLPS) of the host protein LARP7, forming condensates that sequester P-TEFb and the viral Tat protein, thereby inhibiting viral transcription. They argue that LARP7 and Tat co-condense in vitro and in cellulo (the term "in vivo" is incorrect and should be avoided), with LARP7 intrinsically disordered region (IDR) containing 49 lysines being essential for this phase separation. The authors suggest that targeting the phase separation properties of LARP7 could offer new anti-HIV-1 therapeutic strategies. However, they do not investigate how viral infection is subsequently established (LARP7/Tat ratio) and, more interestingly, how this proposed mechanism would affect viral transcription during latency reactivation.

Response:

We appreciate the referee's comments and constructive suggestions on our work. We have followed those comments/suggestions to add more data and in-depth discussions in our revised manuscript.

We thank the referee for pointing out the incorrect use of the term '*in vivo*', and we have replaced it with '*in cellulo*' in the revised manuscript.

The referee pointed out that we do not investigate how viral infection is subsequently established (LARP7/Tat ratio) and, more interestingly, how this proposed mechanism would affect viral transcription during latency reactivation. We agree that how LARP7/Tat ratio dictates the viral infection is very important.

We transiently knocked down LARP7 in Tzm-bl cells and subsequently expressed varying levels of Tat, the cells were then infected with HIV-1 for 24 hours and assessed for viral replication. We found that under certain circumstance, the LARP7/Tat ratio higher, the lower and slower the HIV-1 transcription will be. The data has been incorporated in Figure EV3C and 3D, and corresponding revision has been made in result section.

Figure for referee with unpublished data and its description has been removed upon request by the authors.

We also agree that it is more important that HIV-1-induced LARP7 LLPS would affect viral transcription during latency reactivation, however, is beyond our ability to establish the latency reactivation model and acquire reliable data for now. Additional discussion on our hypothesis and more references were provided in our revised manuscript.

The work would benefit from studies in primary cells and with wild-type virus infection.

Response:

We thank the referee for the suggestion of studies in primary cells and performed experiments accordingly. Primary CD4⁺ T cells derived from human PBMCs were subjected to HIV-1 R7-ΔEnv-GFP virus infection, and we found that the number and fluorescence intensity of LARP7 puncta were increased significantly 12 hours after infection, maintained until 24 hours and declined into homogeneity 48 hours of infection (revised manuscript, Figure1D and E).

Figure for referee with unpublished data and its description has been removed upon request by the authors.

We agree with the referee that our work will be benefit from studies using wildtype HIV-1 virus. However, we have not got access to the facility allowing wildtype virus infection, therefore are not able to get result for this revision.

The text is poorly edited, and figures (especially EVs) are often not fully commented in the results section. Controls are frequently missing, and in general the work is not of sufficient quality to be published in its current state.

Response:

We thank the referee for pointing out the bad editing and missing controls, so we present

additional controls in Figures EV1C, EV1D, EV1E and EV1I etc., and have our final manuscript edited by an English native speaker and hope the referee find it satisfactory.

Major Points:

Figure 1 "After HIV-1 infection, there was a notable rise in the number of LARP7 puncta that had fluorescence intensity higher than those of their surroundings." Since the authors do not show the mean fluorescence intensity of LARP7 puncta but only their number, this sentence is not correct and has to be modified. Also, it is not clear how was this comparison, reported in Fig1c performed? How do the authors know which cells are HIV-1 infected? In the population of cells that they infected and assayed, not all cells (presumably) have HIV-1. It is unclear if the HIV-1 infected cells were compared to surrounding cells?

Response:

We thank the referee for suggesting us to show the average fluorescence intensity of LARP7 puncta together with their number, and we add the data in revised Figure 1B and 1E. We modified the sentence as “Following HIV-1 infection of cells, LARP7 puncta, LARP7 staining signals displayed a markedly higher fluorescence intensity than the surrounding areas, were significantly induced both in number and their fluorescence intensity. These LARP7 puncta persisted for approximately 24 h post-infection and then began to transition to a more homogeneous morphology (Figures 1A and 1B).”

Puncta were identified and counted by Image J software, after all images were standardized with the same “Brightness/Contrast” range (6000-20000) and converted to an 8-bit format, threshold for puncta identification was set to 160 and 255. For the comparison of numbers and intensities of the puncta, we randomly picked 50 cells in some representative images to collect data and analyze statistically between groups. We added descriptions on comparing the fluorescence intensity and the numbers of the puncta in Methods.

Figure for referee with unpublished data and its description has been removed upon request by the authors.

We understand the referee’s concern that without GFP fluorescence at 12 hours after

infection, the exact infection status of the cells that we showed with increased LARP7 puncta cannot be absolutely definite. However, based on the fact that about 75% of the Jurkat cells were GFP positive as we observed in this experiment as well as almost every batch of cells infected after 24-96 hours. Besides, some cells may be infected with HIV-1 but will not express GFP due to the host transcriptional regulation, and may enter latency. So, it's reasonable to assume that at 12 hours, maybe more than 75% of cells are infected with HIV-1.

Figure for referee with unpublished data and its description has been removed upon request by the authors.

In addition, in Jurkat cells treated with mock media and infected with HSV and VSV, cells with increased LARP7 puncta were rare. So, the change of LARP7 puncta shown in Figure 1A is most likely due to HIV-1 infection.

In Figure 1D at 96 hours, when infection peaks, LARP7 levels seem lower compared to the 0-hour time point. Could the authors provide densitometry of the WB replicates? It would be informative to see the same analysis (WB and quantification) performed in Jurkat cells and compared to IF with LARP7 staining in GFP- and GFP+ cells).

Response:

We thank the referee for pointing out that LARP7 levels in 96 hours after HIV-1 infection seem lower compared to the 0-hour time point and suggesting densitometry analysis. Following the referee's suggestion, we performed densitometry analysis on western blotting duplicates from independent infection experiments, and as shown in revised Figures 1C and EV1D, there was no significant difference on LARP7 expression level after infection. Same analysis was performed in CD4⁺ T cells and Tzm-bl cells, and the results are shown in Figures 1F, EV1F and EN1I, and there is no change in the amount of LARP7 after HIV-1 infection.

Figure for referee with unpublished data and its description has been removed upon request by the authors.

At what time point of infection are 1,6-hexanediol (HD) treatments added?

Response:

We apologize for failing to indicate the timepoint for 1,6-hexanediol (HD) treatments. We treated Jurkat cells infected by HIV-1 for 12 h with 1,6-HD or 2,5-HD then fixed the cells with 4% PFA. The information has been provided in revised Figure 1G and corresponding Figure legend.

Clarification needed for the correct interpretation of this OPTOIDR system. It seems that the authors somehow do not fully understand the principle: "To further investigate the LLPS features of LARP7, we adopted a photo-induced phase separation research system, in which a photo-sensitive protein, Cry2, was fused with mCherry-LARP7. When blue light is on, mCherry-LARP7 molecules are brought closer to each other (Figure 1G) (Shin, Berry et al., 2017)."

Response:

We have conducted a thorough review of the OPTOIDR system and revised the description in the article. The updated text is as follows:

To further investigate the phase separation features of LARP7, we adopted a photo-induced phase separation research system. Cry2 is a light-sensitive protein which is

known to self-associate upon blue light exposure. When fused with a phase-separating protein, specifically an IDR containing protein or fragment, Cry2 will mediate light-dependent phase separation of the fusion protein in cell (Shin et al, 2017). As shown in Figure 1L, Cry2-mCherry-LARP7 formed spherical droplets under blue light, and droplets fusion was observed, exhibiting typical liquid-like properties. By contrast, Cry2-mCherry alone did not form droplets after blue light stimulation (Figure EV2B). These results suggest that the formation of the Cry2-mCherry-LARP7 droplets was driven by phase separation of LARP7. Additionally, fluorescence recovery after photobleaching (FRAP) analysis indicated that these droplets undergo dynamic exchange with the surrounding dilute phase (Figures 1M and 1N).

The authors use inconsistent concentrations of 1,6-HD.

Response:

The inconsistency on concentrations of 1,6-HD is just a typo and we sincerely apologize for not proof-reading our text carefully enough. The 1,6-HD concentration we used in all the experiments we presented in the Figures is 6%.

Concluding sentence: "Taken together, the results above suggest that LARP7 undergoes LLPS both in vivo and in vitro, and the phase separation of LARP7 is induced when cells were infected by HIV-1." should be corrected to "in cellulo."

Response:

We thank the suggestion. All the “*in vivo*” in our manuscript have been replaced with “*in cellulo*”.

The term "partition coefficient (EV _J-N)" is mentioned without explanation in the text.

Response:

We thank the reviewer for pointing that out. Partition coefficient serves as an indicator of the extent of protein phase separation (Banani *et al*, 2016). It was calculated as the total fluorescence intensity of droplets divided by the bulk fluorescence intensity of background. Consequently, it serves as an indicator of the extent of protein phase separation. We added the description of the partition coefficient to the results section, figure legends and methods.

Figure 2:

The first sentence is incomplete: "As a core component of the 7sk snRNP, LARP7 maintains the stability of the complex, which prevents P-TEFb from activating (Krueger et al., 2008)."

Response :

We apologize for the lack of clarity in this description. We rewrote the sentence and revised the manuscript: "LARP7, a core component of the 7SK snRNP complex, stabilizes the complex, enabling 7SK snRNP to bind to P-TEFb and maintain it in an inactive state, thereby inhibiting transcription (Krueger et al., 2008)."

Figure 2B-2D: It is not clear if P-TEFb is entirely sequestered within LARP7 condensates, or if there is a separate pool of P-TEFb that interacts with Tat and not with LARP7. Therefore, a triple staining would be more appropriate to assess co-condensates of Tat, LARP7, and Cyclin T1.

Response :

We thank the referee for the constructive suggestion. Unfortunately, after testing multiple combination of antibodies from different providers, and we added the HA-tag to Tat on the HIV-1 R7- Δ Env-GFP plasmid, but specific immunofluorescence staining remained unsuccessful. We still couldn't get a triple staining with satisfactory specificity and resolution.

According to our results in Figures 2A and 2C, some of the puncta of Cyclin T1 and Tat were not co-located with LARP7, and GFP could be observed at this time, which may be due to that some Tat had broken away from the sequestering of LARP7 and started HIV-1 transcription together with active P-TEFb.

Figure 2E: Condensates in HEK cells have a completely different morphology and are very large and few, unlike observations from HIV-1 relevant cells, Jurkat.

Response :

We thank the referees for pushing us to clarify a confusion that have been bewildering us. In our hand, the morphology of overexpressed LAPR7 in cells both in HEK293T cells and Tzm-bl cells, have been always very large and few, we simply thought it's because of overexpression, until we get the same comments from all three referees that indicate the large structure could be nucleoli. According to the referees' suggestion, we performed nucleolus staining using an anti-NPM1(B23) antibody while investigating the morphology of LARP7

and Tat. As shown in revised Figures EV3A, mCherry-LARP7 co-condensates with mEGFP-Tat in nucleoli.

Figure for referee with unpublished data and its description has been removed upon request by the authors.

Though looks very differently, LARP7 puncta can be easily found resided in the nucleolus of HIV-1 infected CD4⁺ T cells. The expression level and the different composition of interacting molecules, including proteins and RNAs in different cells, might be the reason that LARP7 concentration in nucleolus is much more pronounced in transfected HEK293T cells.

Figure for referee with unpublished data and its description has been removed upon request by the authors.

Figure 3

How were the defined lysine sets selected for the mutation experiments - it is not clear on what basis have the authors selected 30, 21 and 13 lysine residues (out of 49) for the downstream analysis.

Response :

We thank the referee very much for the question. Many IDRs have a biased amino acid composition and may be repetitive in sequence, and the conserved, repeated and continuous

lysine in LARP7 IDR stood out the most. So, when reducing the mutated lysine residues, we prioritized the mutation of more contiguous and conserved lysine residues. We added this explanation in the revised manuscript:

“To investigate the role of these lysine residues in the LARP7 IDR in LLPS, variants were constructed by mutation of selected lysine residues to alanine. We first mutated 49 of the 51 lysine residues in LARP7 IDR simultaneously to alanine (retaining two lysine residues within the nuclear localization sequence), and then we reduced the mutated lysine residues and constructed the mutant 30KA (mutated the most contiguous and conserved 30 lysine residues), 21KA (recover left conserved lysine residues from 30KA in 195aa-230aa and 274-314aa) and 13KA (only the most contiguous lysine residues were mutated, 218-225aa and 274-278aa) (Figure 3D).”

Panel F: The staining's are really strange. To me those look-like nucleoli rather than LLPS structures. The authors should use the B23 (nucleophosmin) or nucleolin to verify this. Moreover, these structures do not correlate to what is shown in Figure 2.

Response :

We thank the referee very much for pointing that out. According to the referees' suggestion, we performed nucleolus staining while investigating the morphology of LARP7 and mutants. As shown in revised Figure EV3G and EV3F, when overexpressed in HEK293T cells, mCherry-LARP7 forms condensates within nucleoli, the 49KA mutant abolished both condensation and nucleolus localization of mCherry-LARP7, 30KA mutant localized evenly in both nucleoli and nucleoplasm, no condensates formed, 21KA co-condensated with Tat in nucleoli, 13KA presented co-condensate with Tat in nucleoli and presented similar morphology as mCherry-LARP7 WT.

Figure for referee with unpublished data and its description has been removed upon request by the authors.

Additionally, we used the OptoIDR system to validate the LLPS of LARP7 mutants, and

the results showed that 49KA and 30KA could not form droplets upon blue light, 21KA could partially recover the LLPS of LARP7, and 13KA displayed the same behavior of wild type LARP7. So, the mutant of lysine residues in IDR could disrupt both the nucleolus localization and LLPS of LARP7. The results were added in revised Figure 4E.

Figure for referee with unpublished data and its description has been removed upon request by the authors.

It is a consensus that the nucleolus is also LLPS structure, representing a multilayered biomolecular condensate (Lafontaine *et al*, 2021). It is interesting that the morphology of LARP7 is completely different in HEK293T cells. The expression level and the composition of interacting molecules, including proteins and RNAs in HEK293T cell nuclear might be the reason that overexpressing LARP7 is incorporated in nucleolus.

Figure 4

All immunoblots lack molecular weight markers.

Response:

We thank the referee for the point, and put molecular weight markers in all immunoblots.

Clarify Tat transfection versus cell infection and luciferase readings.

Response:

We thank the referee for the point. When transfected, Tat would be expressed more quickly and stably, in contrast, when cells are infected, Tat expression would go through multiple regulatory steps. Therefore, transfection experiment could more directly observe the effect LARP7 and its mutants on the Tat activated luciferase reading. While, viral infection is more closely related to physiological conditions.

Tat was transfected to activate the reporter and Renilla luciferase reporter plasmid

(pRL-TK) was transfected to normalize the transfection efficiency. This information has been put in the figure legend of Figure 4A in revised manuscript.

Figure 4B: The FLAG immunoblot is missing.

Figures 4B, C: What is the second band in the LARP7 WB?

Figure 4C: The FLAG immunoblot is missing.

Response:

We thank the referee for the point. LARP7 antibody was used in Western blotting experiments, therefore both endogenous LARP7, 3×Flag-LARP7 and mutants were detected. The upper bands are 3×Flag-LARP7, and the lower bands are endogenous LARP7.

Figure 4D: The LARP7 immunoblot is missing.

Response:

We thank the referee for the point. We apologize for a misleading indication in the Figure. The Flag, should be 3×Flag -LARP7. The correction has been made in revised Figures.

Reverse IPs are needed to confirm that IDR lysine residues are crucial for the interaction between LARP7 and components of P-TEFb.

Response:

We thank the referee for the constructive suggestion. Reverse immunoprecipitation experiment has been performed. HA-CDK9 and Flag-LARP7 (Wt/mut) were transfected in HEK293T cells, anti-HA antibody was used to immunoprecipitate the interacting proteins. Cyclin T1 and HEXIM1 as well as Flag-LARP7 mutants were detected using Western blotting. The results showed that with equal amount of HA-CDK9 has been immunoprecipitated, equal amount of Cyclin T1 and HEXIM had been co-precipitated, regardless of whether wild type LARP7 or mutant was presented. LARP7 WT could be co-precipitated with HA-CDK9, similar amount of 21KA and 13KA could be precipitated ,30KA could also be precipitated but the amount was significantly decreased, and hardly any LARP7 49KA could be co-precipitated. These results suggest that the mutation of lysine residues in the IDR interrupts the interactions between LARP7 and P-TEFb components.

Figure for referee with unpublished data and its description has been removed upon request by the authors.

Referee 2#

Li et al characterize how LARP7 undergoes LLPS in HIV-1 infected cells and how this affects viral replication. They show that LARP7 puncta are induced upon HIV-1 infection. Li et al then propose that these puncta are formed via LLPS because their abundance is decreased when cells are treated with 1,6-Hexanediol. They also show that the LARP7 puncta can fuse and have liquid-like properties in FRAP experiments. Cyclin T1 and HIV-1 Tat were also in LARP7 puncta. In addition, Tat and LARP7 formed droplets in vitro. Therefore, the authors have shown that Tat, Cyclin T and LARP7 can form LLPS-like puncta.

A difficult challenge for many proposed LLPS systems is showing their functional relevance in a living cell. The authors have attempted to address this by mutating the large number of lysines in the LARP7 linker region between the La module and RRM2. They mutated 49, 30, 21 or 13 lysine residues to alanine and analyzed whether this affects LARP7 puncta formation or HIV-1 transcription. The role for these lysine residues in LARP7 function is not clear, though many appear to be evolutionarily conserved. The 49KA or 30KA mutants did not form puncta and did not inhibit HIV-1 gene expression as efficiently as wild type LARP7. Proteins with 21 or 13 lysines mutated had activity more similar to the wild type protein. The 49KA and 30KA mutant proteins had reduced interaction with CDK9 and Cyclin T1, leading to the question of whether this is a specific effect on LLPS or whether the mutant proteins have lost several functions due to the mutations in the conserved linker region (see below).

Response:

We thank the referee very much for his or her comments and constructive suggestions. We performed additional experiments to clarify the concerns, the following are point-to-point responses to the referee's comments.

Major comments:

1. The major point of interpretation is whether the lysines in the linker region have a function for LARP7 independent of LLPS and how this could affect its function in regulating

transcription. The authors should show whether mutating these lysines have affected LARP7 binding to 7SK RNA. Previous reports have found that the basic residues in the central linker of LARP7 may contribute to 7SK binding (reviewed in RNA Biol. 2021;18:290-303), so these mutations may have a global effect on LARP7 activity that is independent from LLPS. The authors should also analyse whether these mutations have decreased interaction with MePCE. Based on the decreased interaction with CDK9 and Cyclin T1, it is possible that the lysine mutations have decreased the ability of LARP7 to interact with the 7SK complex due to decreased RNA binding. It is essential to rule out possibilities other than LLPS formation that the conserved linker region may contribute to.

Response:

We thank the referee very much for the suggestion. We investigated the effect of lysine residue mutations on the interaction between LARP7 and 7SK snRNA or MePCE using a RIP-qPCR assay. Flag-LARP7 and mutants were expressed in HEK293T cells, and isolated using anti-Flag beads. RNAs co-precipitated were then extracted by using TRI reagent, and the level of 7SK snRNA were detected by RT-qPCR. As shown in revised Figure 4E, only 49KA mutant failed to bind any 7SK snRNA, the 30KA, 21KA and 13KA mutants bound to similar level of 7SK snRNA with wild type LARP7.

The interaction of LARP7 and mutants with MePCE and HEXIM1 were analyzed using Co-IP assay. As shown in revised Figure 4F, the interactions between both 7SK snRNP components and P-TEFb components were almost completely disrupted by 49KA mutation, 30KA mutation led to significantly reduced interactions from P-TEFb components and HEXIM1, but not MePCE, 21KA and 13KA mutants bound to both complex almost as well as WT.

Taken together, 30KA mutation breaks LARP7 LLPS without interrupting the interactions between 7SK snRNA and 7SK snRNP component MEPCE, and only partially interferes the HEXIM1 and P-TEFb binding, while almost completely abolishes LARP7 inhibition on Tat dependent HIV-1 transcription, suggesting LLPS is indispensable for LARP7 to inhibit HIV-1 transcription. However, the contribution of the conserved linker region could not be completely ruled out and need more work to clarify.

Figure for referee with unpublished data and its description has been removed upon request by the authors.

(

2. *Figure 4C needs to be quantified in multiple independent experiments. The magnitude of the effect of LARP7 on HIV-1 transcription appears to be small (3-fold in Figure 4B, maybe less in Figure 4C). How important is this phenotype for HIV-1 transcription? Measuring viral replication using full length HIV-1 would be more sensitive than the GFP or luciferase reporter experiments.*

Response:

We thank the referee very much for the suggestion. Quantification was performed in multiple independent experiments, and the result was shown in revised Figure EV4D. Additional readouts, including the percentage of the GFP positive cells, mRNA levels of HIV-1 *ltr* and *pol*, were also measured, and the magnitudes of effect of LARP7 on HIV-1 transcription are between 2.5 to 8-fold.

The limited transfection efficiency and incomplete knockdown of endogenous LARP7 might confine the magnitude of the effect.

Figures for referee with unpublished data and its description has been removed upon request by the authors.

Figure for referee with unpublished data and its description has been removed upon request by the authors.

Nanopore sequencing were employed to analyze the HIV-1 transcripts in both control and LARP7 knockdown cells. Total RNAs were extracted from the cells, then sequencing library were prepared using SQK-LSK110 kit (Oxford Nanopore Technologies) and sequenced with a PromethION R9.4.1 (FLO-PRO002) flowcell. The resulting sequence were aligned to HIV-1 genome using CLC Genomics Workbench 23 (Qiagen). As shown here, 16 reads totaling 10,777 bases for the control group and 45 reads amounting to 52,370 bases to for LARP7 knockdown group were mapped to HIV-1 genome, and the sequence coverage and depth are both much higher in LARP7 knockdown group than that in control group. These results suggest that longer and more complete HIV-1 transcripts would be produced without LARP7.

Figure for referee with unpublished data and its description has been removed upon request by the authors.

3. The model proposed by the authors needs to be refined. Tat forms a positive regulatory circuit in which low levels of full-length HIV-1 transcription lead to a small pool of Tat which then promotes more efficient transcription and increased Tat levels and therefore high levels of processive HIV-1 transcription. The authors propose at the end of the results section that early in HIV-1 infection, low levels of Tat is trapped with P-TEFb in LARP7 droplets, which would make it inactive. This is hard to reconcile with the positive regulatory model in which low levels of Tat induce viral transcription. The role of Tat in regulating HIV-1 transcription is complex and there is no analysis of how it binds to HIV-1 RNA or leads to Pol II

phosphorylation in the experimental conditions in this manuscript. It is also not clear how HIV-1 infection leads to the induction of LARP7 foci. Without more information, the model is currently incomplete and it is hard to determine the functional role of LARP7 puncta based only on the lysine mutations in LARP7.

Response:

We thank the referee for this important point. As mentioned by the referee, a well-accepted model of Tat transactivation is that Tat forms a positive regulatory circuit in which low levels of full-length HIV-1 transcription leading to a small pool of Tat which then promotes more efficient transcription and increased Tat levels and therefore high levels of processive HIV-1 transcription. However, we found that Tat expression need to reach certain level to initiate the positive regulatory circuit, and the Tat threshold could be regulated by LARP7. This was supported by the following experiments (also in revised Figure EV3C and EV3D.)

We transiently knocked down LARP7 in Tzm-bl cells and subsequently expressed varying levels of Tat, the cells were then infected with HIV-1 for 24 hours and assessed for viral replication. The results indicated that when LARP7 was present, the pre-expressed Tat cannot further enhance HIV-1 transcription. In contrast, when LARP7 was absent, there was a significant increase in HIV-1 transcription, and it was further enhanced with the rise in the amount of pre-expressed Tat. These findings suggested that LARP7 plays a crucial role in regulating Tat-mediated HIV-1 transcription.

Figure for referee with unpublished data and its description has been removed upon request by the authors.

We further analyzed the co-localization of phosphorylated RNAPII with LARP7 and with Tat respectively using immunofluorescence in Jurkat cells 24 hours after HIV-1 infection. As shown here, LARP7 droplets, with and without co-concentrated p-RNAPII could be detected in the same cell. Similar phenomenon was observed for Tat and p-RNAPII. Given the fact that virus transcription is activated in these GFP-positive cells, Tat and p-RNAPII co-localize could be explained. However, we currently couldn't understand why LARP7 and p-RNAPII co-localize. Unfortunately, after testing multiple combination of antibodies from different providers, specific immunofluorescence staining remained unsuccessful. We still couldn't get a triple staining with satisfactory specificity and resolution.

We agree with the referee that more evidence is still needed to fully support our model, so we put it in revised Figure EV4D to help illustrating our hypothesis.

Figure for referee with unpublished data and its description has been removed upon request by the authors.

To investigate how HIV-1 infection leads to the induction of LARP7 foci, Jurkat cells were infected with R7- Δ Env-GFP or R7- Δ Tat- Δ Env-GFP virus. R7- Δ Tat- Δ Env-GFP virus infection hardly induced any changes in LARP7 morphology, while the efficiency is similar with that of R7- Δ Env-GFP confirmed using qPCR on the integrated HIV-1 sequence in Jurkat cell genome, suggesting that LARP7 puncta were triggered by Tat-dependent process of HIV-1 infection.

Figure for referee with unpublished data and its description has been removed upon request by the authors.

Referee #3:

General comment:

In this manuscript, Li et al. explored the functional interaction between HIV-1 Tat protein and LARP7, a subunit of the 7sk snRNP which sequesters and inhibit PTEFb kinase activity. They show that LARP7 has an intrinsic ability to undergo liquid-liquid phase separation (LLPS). They identified the intrinsically disordered region within and the associated lysine residue responsible for the formation of LARP7 droplets. LARP7 condensates are increased upon HIV-1 infection. The HIV-1 viral transactivator Tat and Cyclin T1, the regulatory subunit of PTEFb, are associated with LARP7 droplets. Finally, they show that LARP7-mediated inhibition of Tat activation of HIV-1 transcription is LARP7 IDR-dependent. The topic is of importance not only to the field of HIV-1 but also in our understanding of PTEFb regulation. However, as it stands, the manuscript suffers from lack of important controls and additional experiments to support the authors conclusions. Additionally, the manuscript needs language and grammar editing.

Response:

We thank the referee for the comments and constructive suggestions. We performed additional experiments and included more controls to further support our conclusions. In addition, we had our manuscript edited on language and grammar by a professional. We hope the referee find our manuscript much improved now.

Major comments:

*1*Figure 1 and EV1: the concern here is lack of controls both for the anti-LARP7 antibody and for the specificity towards HIV-1. Thus, experiment showing the staining with anti-LARP7 in control cells versus cells transfected with LARP7 specific siRNA is needed.*

Response:

We thank the referee for the constructive suggestion.

The specificity of the LARP7 antibody was the first thing we have made sure before we started LARP7 staining experiments. To do so, immunofluorescence staining and Western blot assays were performed in on Jurkat cells with stable LARP7 knockdown and control cells. The results were incorporated in revised Figure EV1C. LARP7 can be detected in a clean, single band in Jurkat cells with Western blotting, and the blot is significantly reduced in LARP7 knocked-down cells, the immunofluorescence is significantly weaker in LARP7 knocked-down cells compared to control cells, suggesting that the anti-LARP7 antibody (17067-1-AP, Proteintech) is specific.

Figure for referee with unpublished data and its description has been removed upon request by the authors.

The use of VSV and HSV is highly appreciated. However, use of HIV-1 Tat minus virus or lentiviral vector and viral like particles (VLPs) will be of importance to assess what triggers the formation of LARP7 droplets. Indeed, enhanced LARP7 droplets is observed at 12hrs post infection where Tat has not yet been expressed. Interestingly, the viral protein R (VPR) has been shown to target CTIP2 which associate with and regulates 7SKsnRNP (Eilebrecht S. et al. NAR. 2014).

Response:

We thank the referee for the constructive suggestion. According the referee's suggestion, a R7- Δ Tat- Δ Env-GFP plasmid has been constructed. However, when this plasmid and VSV-g were transfected to HEK 293T cells, no obvious green fluorescence was observed, indicating that virus particles could not be successfully packaged. We then figured out that Tat is also very important for the packaging stage of R7 virus. We finally managed successful packing through co-transfect Tat in a separate plasmid.

Then Jurkat cells were infected with R7- Δ Env-GFP or R7- Δ Tat- Δ Env-GFP virus, LARP7 was detected using IF before and after 24 hours infection. R7- Δ Tat- Δ Env-GFP virus infection hardly induced any changes in LARP7 morphology, while the efficiency is similar with that of R7- Δ Env-GFP confirmed using qPCR on the integrated HIV-1 sequence in Jurkat cell genome, suggesting that LARP7 puncta were triggered by Tat-dependent process of HIV-1 infection.

Figure for referee with unpublished data and its description has been removed upon request by the authors.

In our hand, it is true that we can hardly detected any Tat expression on protein level, neither using WB nor IF 12 hours post infection. However, though very low, GFP can be detected using Western blot (in revised Figure 1C), suggesting low level of virus encoding protein expression is ongoing. We reasoned that there should be low level Tat expression beyond our detection sensitivity.

To find out the possible role the viral protein R (Vpr) might play in HIV-1 induced LARP7 condensations, we first expressed both mCherry-LARP7 and GFP-Vpr in HEK293T cells. As shown here, mCherry-LARP7 and GFP-Vpr exhibited no co-condensation, suggesting that Vpr does not participate in HIV-1 induced LARP7 condensations by directly interacting with LARP7. However, the exact function of Vpr in regulating the LLPS of LARP7 could only be found out using a Vpr deficient virus that we don't have for now.

Figure for referee with unpublished data and its description has been removed upon request by the authors.

*2*Figure 2E, 3F, EV2: The concern here is about the subcellular localization of ectopically expressed LARP7 and Tat. Indeed, both LARP7 and Tat are RNA binding proteins. Their overexpression results in accumulation in RNA enriched organelles such as the nucleolus. Thus, use of Tat and LARP7 mutants lacking RNA binding domains is required.*

Response:

We thank the referee for the constructive suggestion. We constructed truncated mutants of LaM, RRM1, and RRM2, individually or in combination (see below).

To our surprise, none of the truncates showed significant reduction in nucleolus localization. In addition, as we showed in revised Figures 3E, EV3F and EV3G, 49KA and 30KA mutations in the LARP7 IDR abolished the nucleolus localization even with intact LaM, RRM1, and RRM2 that has been reported mediate the interaction with 7SK snRNA. These data suggested that the IDR region is essential for the RNA binding and nucleolus localization, and at the same time, 7SK snRNA binding is not sufficient for LARP7 foci formation, because 30KA mutation forms no foci even though it can bind to 7SK snRNA and is localized in nucleoli.

Figure for referee with unpublished data and its description has been removed upon request by the authors.

Tat mutant lacking RNA binding domain (Δ 48-59aa) was localized in both the cytoplasm and nucleus. The RNA binding domain of Tat is required for its phase separation and nucleolus localization.

Figure for referee with unpublished data and its description has been removed upon request by the authors.

It has been shown that Tat displace hexim1 from the 7SKsnRNP and form a stable complex with 7SKRNA/LARP7/MEPCE through direct interaction with 7SKRNA (Sobhian et al. Mol.Cell. 2010; Muniz L. et al. Plos Pathog. 2010 and others). Thus, one question the presence of HEXIM1 in LARP7 droplets containing Tat and CyclinT1. This should be addressed by assessing the presence of HEXIM1 in LARP7 droplets containing Tat and CyclinT1.

Response:

We thank the referee for the constructive suggestion. We performed a triple staining of LARP7, HEXIM1 and Cyclin T1 in Jurkat cells 24 hours after HIV-1 infection. The results showed that, HEXIM1 presented in the LARP7 droplets containing Cyclin T1, indicating that Tat didn't displace all HEXIM1 from the 7SK snRNP. The 7SK snRNP may induced after

HIV-1 infection to prevent virus replication. We regretted that we could not obtain the result of the triple staining of LARP7, HEXIM1 and Tat, due to the less effectiveness of the anti-Tat antibody.

Figure for referee with unpublished data and its description has been removed upon request by the authors.

CyclinT1 has been shown to undergo LLPS (Lu H. et al. Nature 2018). It will be important to analyze whether TatC22G mutant, unable to bind CyclinT1, localizes to LARP7 droplets containing CyclinT1.

Response:

We thank the referee for the constructive suggestion. We constructed the plasmid of TatC22G and co-expressed it with mCherry or mCherry-LARP7. We found that TatC22G mutant showed no less concentration in nucleolus or co-localization with LARP7, suggesting that Tat interacting with Cyclin T1 is not required for its nucleolus localization.

Figure for referee with unpublished data and its description has been removed upon request by the authors.

*3*Figure 3: Results shown in 3F is of major concern (see comment 2). Indeed, LARP7 and Tat colocalize only when they both concentrate in the nucleolus. The use of LARP7 lacking its RNA binding domain is important. This mutant should not localize to the nucleolus but should still be able to form droplets. Question is whether this mutant is still able to trap Tat into the droplets.*

Response:

We thank the referee for the constructive suggestion. We constructed truncated mutants of LaM, RRM1, and RRM2, individually or in combination. But none of the truncates disrupted the foci formation or nucleolus localization of LARP7 (see also response to the previous point). Previous studies have reported that the central linker of LARP7 (the IDR we predicted), may also contribute to 7SK snRNA binding (Eichhorn *et al*, 2018; Uchikawa *et al*, 2015). The results of OPTO-IDR assay (revised Figure 3E), subcellular localization (revised Figures EV3G and EV3F) and RIP-qPCR assay (revised Figure 4E), showed that 30KA though bind to 7SK snRNA, cannot undergo LLPS and inhibit Tat transactivation, suggesting that RNA binding is not sufficient for LARP7 LLPS. Nonetheless, we haven't been able to develop any strategies to abolish the nucleolus localization but not affect the LLPS of LARP7.

*4*Figure 4: In absence of the above proposed experiment, the functional results shown in figure 4 are hard to interpret. Indeed, as previously shown LARP7 knockdown will lead to 7SK RNA degradation and increased active PTEFb which explains the increased Tat transcriptional activity towards the viral LTR. An important experiment to be performed is to ask whether RNAPII associated with LARP7 droplets containing Tat is phosphorylated at Ser5 or Ser2. This can be performed with specific antibodies.*

Response:

We thank the referee for the constructive suggestion. We further analyzed the co-localization of phosphorylated RNAPII with LARP7 and with Tat respectively using immunofluorescence in Jurkat cells 24 hours after HIV-1 infection. As shown here, LARP7 droplets, with and without co-concentrated p-RNAPII could be detected in the same cell. Similar phenomenon was observed for Tat and p-RNAPII. Given the fact that virus transcription is activated in these GFP-positive cells, Tat and p-RNAPII co-localize could be explained. However, we currently couldn't understand why LARP7 and p-RNAPII co-localize. Unfortunately, after testing multiple combinations of antibodies from different providers, specific immunofluorescence staining remained unsuccessful. We still couldn't get a triple staining with satisfactory specificity and resolution.

Figure for referee with unpublished data and its description has been removed upon request by the authors.

To exclude artificial aggregates, one may consider using different, shorter tags or an otherwise different detection system (even simple tags such as HA) expressed from a weak to moderate promoter (not CMV or SV40 for instance but rather LTR driven or inducible but weakly to moderately induced).

Response:

We thank the referee for the important point. Following the referee's suggestion, Flag tagged LARP7 was stably transfected in Tzm-bl cells. Flag IF was used to examine LARP7 morphology, nucleolus localization is still pronounced, though not as strong as mCherry-LARP7 expressed in HEK293 cells. Besides, LARP7 puncta can be easily found resided within the nucleolus of HIV-1 infected CD4⁺ T cells. We believe that the expression level and the different composition of interacting molecules, including proteins and RNAs in different cells, might be the reason that LARP7 is concentrated in nucleolus in HEK293T cells.

Figure for referee with unpublished data and its description has been removed upon request by the authors.

Introduction

1. « Specifically speaking, when Tat functions normally, HIV-1 enters replication cycle,

whereas when Tat dependent transcription is inhibited, HIV-1 primarily enters dormancy,

which is the main obstacle for HIV-1 eradication (Razooky, Pai et al., 2015).

The HIV-1 promoter contains binding sites for host cell transcription factors such as NFκB and SP1, driving basal levels of transcription. Tat in this system is considered as a positive feedback loop. Latency may thus not be solely due to lack or presence of Tat activity. Please consider the following paper: Jordan et al. The EMBO Journal 22(8):1868-77.

Response:

We thank the referee for the point. We have revised this sentence in the manuscript as “Specifically speaking, Tat initiates a positive feedback loop that greatly promote HIV-1 replication, whereas when this loop is halted, HIV-1 may enter dormancy, which could be reversed by introducing Tat or other transcription activator such as NF-κB and SP1 (Jordan et al, 2003; Razooky et al, 2015).”

2. *"Evidence showed that LARP7 also plays an inhibitive role in HIV-1 replication."*

Please provide a reference

Response:

We apologize for missing the citation. A reference (Markert *et al*, 2008) has been added in the revised manuscript.

3. *"Additionally, targeting the phase separation of LARP7 could potentially serve as a novel approach for treating AIDS."*

It is unclear how LARP7 LLPS properties may be targeted for treating AIDS.

Response:

We thank the referee for the important point. In our study, we suggested that the LLPS of LARP7 could inhibit HIV-1 transcription. Promoting the LARP7 and Tat condensates formation may restrain virus replication. On the other hand, it has been reported that disrupting the formation of the 7SK snRNP complex can assist in reversing HIV-1 latency (Stoszko *et al*, 2020). Thus, we hypothesize that inhibiting LARP7 phase separation might disrupt its inhibition of P-TEFb and Tat, thereby facilitating the release of HIV-1 from latency and the treatment of AIDS in combination with antiretroviral therapy.

Methods

1. *"And Tzm-bl cells were treated with CSK buffer (10mM PIPES PH7.0, 100mM NaCl, 300mM sucrose, 3mM MgCl2, 0.5% Triton X-100) for 5 mins as described previously."*

Please provide a reference.

Response:

We apologize for missing the citation. A reference (Egloff *et al*, 2017) has been added in the revised manuscript method section.

References

- Banani SF, Rice AM, Peeples WB, Lin Y, Jain S, Parker R, Rosen MK (2016) Compositional Control of Phase-Separated Cellular Bodies. *Cell* 166: 651-663
- Egloff S, Vitali P, Tellier M, Raffel R, Murphy S, Kiss T (2017) The 7SK snRNP associates with the little elongation complex to promote snRNA gene expression. *The EMBO journal* 36: 934-948
- Eichhorn CD, Yang Y, Repeta L, Feigon J (2018) Structural basis for recognition of human 7SK long noncoding RNA by the La-related protein Larf7. *Proceedings of the National Academy of Sciences of the United States of America* 115: E6457-e6466
- Jordan A, Bisgrove D, Verdin E (2003) HIV reproducibly establishes a latent infection after acute infection of T cells in vitro. *The EMBO journal* 22: 1868-1877
- Lafontaine DLJ, Riback JA, Bascetin R, Brangwynne CP (2021) The nucleolus as a multiphase liquid condensate. *Nature Reviews Molecular Cell Biology* 22: 165-182
- Markert A, Grimm M, Martinez J, Wiesner J, Meyerhans A, Meyuhas O, Sickmann A, Fischer U (2008) The La - related protein LARP7 is a component of the 7SK ribonucleoprotein and affects transcription of cellular and viral polymerase II genes. *EMBO reports* 9: 569-575
- Razooky B, Pai A, Aull K, Rouzine I, Weinberger L (2015) A hardwired HIV latency program. *Cell* 160: 990-1001
- Stoszko M, Al-Hatmi AMS, Skriba A, Roling M, Ne E, Crespo R, Mueller YM, Najafzadeh MJ, Kang J, Ptackova R *et al* (2020) Gliotoxin, identified from a screen of fungal metabolites, disrupts 7SK snRNP, releases P-TEFb, and reverses HIV-1 latency. *Science advances* 6: eaba6617
- Uchikawa E, Natchiar KS, Han X, Proux F, Roblin P, Zhang E, Durand A, Klaholz BP, Dock-Bregeon AC (2015) Structural insight into the mechanism of stabilization of the 7SK small nuclear RNA by LARP7. *Nucleic Acids Res* 43: 3373-3388

Dear Dr. Li,

Thank you for the submission of your revised manuscript to our editorial offices. I have now received the reports from two of the three referees that were asked to re-evaluate the study, you will find below. As you will see, both referees now support the publication of the study in EMBO reports. Referee #3 agreed to look into the revision, but did not submit a report, despite several reminders. However, referee #2 assessed your p-b-p-response and indicated that the points of referee #3 have also been adequately addressed during revision. Referee #1 has remaining concerns and suggestions to improve the study, I ask you to address in a final revised manuscript. Please also provide a final p-b-p-response addressing the remaining points of the referee.

Moreover, I have these editorial requests I also ask you to address:

- We plan to publish your manuscript as Report, as there are only 4 main figures. For a Scientific Report we require that results and discussion sections are combined in a single chapter called "Results & Discussion". Please do this for your manuscript.

- Please add the authors affiliations to the title page of the manuscript main text file.

- Please provide individual production quality figure files as .eps, .tif, .jpg (one file per figure), of main figures and EV figures. Please upload these as separate, individual files upon re-submission.

- We updated our journal's competing interests policy in January 2022 and request authors to consider both actual and perceived competing interests. Please review the policy <https://www.embopress.org/competing-interests> and update your competing interests if necessary. Please name this section 'Disclosure and Competing Interests Statement' and put it after the Acknowledgements section.

- Please order the sections like this, using these names:

Title page - Abstract - Keywords - Introduction - Results & Discussion - Methods - Data availability section - Acknowledgements (including the funding information) - Disclosure and Competing Interests Statement - References - Figure legends - Expanded View Figure legends

- We now use CRediT to specify the contributions of each author in the journal submission system. CRediT replaces the author contribution section. Please use the free text box to provide more detailed descriptions and do NOT provide your final manuscript text file with an author contributions section. See also our guide to authors: <https://www.embopress.org/page/journal/14693178/authorguide#authorshipguidelines>

- Please make sure that all figure panels are called out separately and sequentially. Presently, there is no callout for panel 1K. Please check.

- Please check again that the number "n" for how many independent experiments were performed, their nature (biological versus technical replicates), the bars and error bars (e.g. SEM, SD) and the test used to calculate p-values is indicated in the respective figure legends. Please also check that all the p-values are explained in the legend, and that these fit to those shown in the figure. Please provide statistical testing where applicable. Please avoid the phrase 'independent experiment', but clearly state if these were biological or technical replicates. Please also indicate (e.g. with n.s.) if testing was performed, but the differences are not significant. In case n=2, please show the data as separate datapoints without error bars and statistics. See also:

<http://www.embopress.org/page/journal/14693178/authorguide#statisticalanalysis>

If n<5, please show single datapoints for diagrams. Moreover:

- Please provide the exact p values in the legends of figures 1B, E, H; 4A, B, D, E; EV3 C; EV4 D.

- Although 'n' is provided, please describe the nature of entity for 'n' in the legends of figures 2E, 4A, B, D, E; EV1 D, F, I; EV3C, EV4 D.

- Please define the scale bars for figures 1I J, EV1 C

- Please note that the white dotted line is not defined in the legends of figures 1A, D. This needs to be rectified.

- Please note that the red dotted line is not defined in the legend of figure 2A. This needs to be rectified.

- Please note that the yellow, black dotted lines are not defined in the legend of figure 2C. This needs to be rectified.

- Please remove the reagents and tools table from the manuscript text file. Please upload the table separately as 'Reagents and Tools Table' and add callouts to the table in the Methods section where appropriate. More information on how to adhere to this format as well as downloadable templates (.doc) for the Reagents and Tools Table can be found in our author guidelines (section 'Structured Methods'):

- Please add the primer information (Table 2) directly to the 'Reagents and Tools Table'.
- Please add a legend for Table 1 on the first TAB of the excel file.
- Please name the movie file 'Movie EV1' and use this name for its file name and callouts. Moreover, please provide a legend as a readme.txt file and upload it ZIPed together with the movie file.

In addition, I would need from you uploaded separately:

Best,

Referee #1:

The authors have made significant efforts to address the comments which I and other reviewers raised. There are still some parts of the manuscript that require editing (listed below).

One additional remark that I have relates to the amount of transfected Tat-GFP plasmid (Figure 2 and EV3 and shown in rebuttal letter). The amounts of transfected Tat protein are too high, and all of Tat will therefore end up in the nucleolus. As such this experiment is too artificial. Tat can be functional in transcribing the LTR in TZM-b cells in amounts as low as 10 or 20 ng.

Page 3 please check the following sentence:

Following HIV-1 infection of cells, LARP7 puncta, LARP7 staining signals displayed a markedly higher fluorescence intensity than the surrounding areas, were significantly induced both in number and their fluorescence intensity.

Page 4-5: Sentence needs to be corrected: We found that under certain circumstance, the LARP7/Tat ratio higher, the lower and slower the HIV-1 transcription will be.

Page 5: Sentence should be corrected: These results suggest that LARP7/Tat ratio regulate their co-phase separation, we thus wondered whether it can also regulate their function on HIV-1transcription.

Referee #2:

Li et al have revised the manuscript and the new data adds important details. In particular, the RIP experiments in Figure 4E and the co-IP experiments in Figure 4F add important details that the lysine mutations not only affect condensate formation but also affect binding to the components of the complex. The revisions have satisfied my major comments.

Response for editor's requests:

Moreover, I have these editorial requests I also ask you to address:

- We plan to publish your manuscript as Report, as there are only 4 main figures. For a Scientific Report we require that results and discussion sections are combined in a single chapter called "Results & Discussion". Please do this for your manuscript.

- Please add the authors affiliations to the title page of the manuscript main text file.

Response:

Thank you for considering publishing our manuscript. We have ordered the results and discussion sections to be a single chapter called "Results & Discussion".

We have added the authors affiliations to the title page of our manuscript main text file.

- Please provide individual production quality figure files as .eps, .tif, .jpg (one file per figure), of main figures and EV figures. Please upload these as separate, individual files upon re-submission.

Response:

We have upload individual production quality figure files as separate, individual files.

- We updated our journal's competing interests policy in January 2022 and request authors to consider both actual and perceived competing interests. Please review the policy

<https://www.embopress.org/competing-interests> and update your competing interests if necessary.

Please name this section 'Disclosure and Competing Interests Statement' and put it after the Acknowledgements section.

Response:

We have added the section 'Disclosure and Competing Interests Statement' after the 'Acknowledgements' section.

- Please order the sections like this, using these names:

Title page - Abstract - Keywords - Introduction - Results & Discussion - Methods - Data availability section - Acknowledgements (including the funding information) - Disclosure and Competing Interests Statement - References - Figure legends - Expanded View Figure legends

Response:

We have ordered the sections according to your suggestions.

- We now use CRediT to specify the contributions of each author in the journal submission system.

CRediT replaces the author contribution section. Please use the free text box to provide more detailed descriptions and do NOT provide your final manuscript text file with an author contributions section.

See also our guide to authors:

Response:

We have added a detailed description of each author's contribution on the journal submission system.

- Please make sure that all figure panels are called out separately and sequentially. Presently, there is no callout for panel 1K. Please check.

Response:

We apologize for this and we have checked the manuscript to make sure all figure panels are called out separately and sequentially.

- Please check again that the number "n" for how many independent experiments were performed, their nature (biological versus technical replicates), the bars and error bars (e.g. SEM, SD) and the test used to calculate p-values is indicated in the respective figure legends. Please also check that all the p-values are explained in the legend, and that these fit to those shown in the figure. Please provide statistical testing where applicable. Please avoid the phrase 'independent experiment', but clearly state if these were biological or technical replicates. Please also indicate (e.g. with n.s.) if testing was performed, but the differences are not significant. In case n=2, please show the data as separate datapoints without error bars and statistics. See also:

<http://www.embopress.org/page/journal/14693178/authorguide#statisticalanalysis>

Response:

We have checked that the number "n" for how many independent experiments were performed, their nature, the bars and error bars and the test used to calculate p-values is indicated in the respective figure legends.

We have checked that test used to calculate p-values is indicated in the respective figure legends and Figures.

We have amended the phrase 'independent experiment' and added the description of the nature of entity for "n", biological or technical replicates.

If n<5, please show single datapoints for diagrams. Moreover:

- Please provide the exact p values in the legends of figures 1B, E, H; 4A, B, D, E; EV3 C; EV4 D.

- Although 'n' is provided, please describe the nature of entity for 'n' in the legends of figures 2E, 4A, B, D, E; EV1 D, F, I; EV3C, EV4 D.

- Please define the scale bars for figures 11 J, EV1 C

- Please note that the white dotted line is not defined in the legends of figures 1A, D. This needs to be rectified.

- Please note that the red dotted line is not defined in the legend of figure 2A. This needs to be rectified.

- Please note that the yellow, black dotted lines are not defined in the legend of figure 2C. This needs to be rectified.

Response:

We have shown all datapoints by a single point for diagrams when $n < 5$.

To improve clarity, we have replaced the asterisks (*) with the exact p-values in the diagrams.

We have described the nature of entity for "n" in the legends of figures 2E, 4A, B, D, E; EV1 D, F, I; EV3C, EV4 D.

We have defined the scale bars for figures 1I, 1J and EV 1C.

We have defined the dotted lines in figures 1A, D and figures 2A, C.

- Please remove the reagents and tools table from the manuscript text file. Please upload the table separately as 'Reagents and Tools Table' and add callouts to the table in the Methods section where appropriate. More information on how to adhere to this format as well as downloadable templates (.doc) for the Reagents and Tools Table can be found in our author guidelines (section 'Structured Methods'):

Response:

We have removed the reagents and tools table from the manuscript text file and uploaded the table separately as 'Reagents and Tools Table' and added callouts to the table in the Methods section where appropriate.

- Please add the primer information (Table 2) directly to the 'Reagents and Tools Table'.

Response:

We have added the primer information (Table 2) directly to the 'Reagents and Tools Table'.

- Please add a legend for Table 1 on the first TAB of the excel file.

Response:

We have added a legend for Table 1 on the first TAB of the excel file.

- Please name the movie file 'Movie EV1' and use this name for its file name and callouts. Moreover, please provide a legend as a readme.txt file and upload it ZIPed together with the movie file.

Response:

We have named the movie file 'Movie EV1' and provided a legend as a readme.txt file and uploaded them together ZIPed.

In addition, I would need from you uploaded separately:

- a short, two-sentence summary of the manuscript (not more than 35 words).

- two to four short (!) bullet points highlighting the key findings of your study (two lines each).

- a schematic summary figure as separate file that provides a sketch of the major findings (not a data

image) in jpeg or tiff format (with the exact width of 550 pixels and a height of not more than 400 pixels) that can be used as a visual synopsis on our website.

Response:

We have uploaded these files separately.

- a short, two-sentence summary of the manuscript (not more than 35 words).

The study reveals that HIV-1 infection induces liquid-liquid phase separation of LARP7, forming condensates that sequester Tat and P-TEFb to inhibit viral transcription. Targeting LARP7 phase separation could offer a novel strategy for combating HIV-1.

- two to four short (!) bullet points highlighting the key findings of your study (two lines each).

1. HIV-1 infection induces liquid-liquid phase separation of LARP7, forming condensates that sequester P-TEFb and Tat, inhibiting viral transcription.
2. LARP7's intrinsically disordered region, particularly conserved lysine residues, is essential for phase separation and its ability to restrict Tat-mediated HIV-1 replication.
3. Targeting LARP7 phase separation could be a novel strategy for combating HIV-1, offering new therapeutic insights for viral infections and related diseases.

- a schematic summary figure as separate file that provides a sketch of the major findings (not a data image) in jpeg or tiff format (with the exact width of 550 pixels and a height of not more than 400 pixels) that can be used as a visual synopsis on our website.

HIV-1 induced phase separation of LARP7

Response for Referees' comments:

Referee #1:

The authors have made significant efforts to address the comments which I and other reviewers raised. There are still some parts of the manuscript that require editing (listed below).

One additional remark that I have relates to the amount of transfected Tat-GFP plasmid (Figure 2 and EV3 and shown in rebutal letter). The amounts of transfected Tat protein are too high, and all of Tat will therefore end up in the nucleolus. As such this experiment is too artificial. Tat can be functional in transcribing the LTR in TzM-b cells in amounts as low as 10 or 20 ng.

Response :

We appreciate the referee's comment regarding the amount of transfected Tat-GFP plasmid used in our experiments. To find out whether the nucleolus localization is due to the artificiality caused by large amount of exogenous protein expression, we reviewed images from 12 h after Tat transfection, when the green fluorescence was just starting to be detectable and was very weak in some cells. As shown here, though very weak, GFP-Tat formed condensates similarly to those depicted in Figure 2 and Figure EV3.

Figure for referee with unpublished data and its description has been removed upon request by the authors.

We attempted to transfect 10-20 ng of plasmid into Tzm-bl cells. However, due to the low transfection efficiency, no green fluorescence was observed, even at 12 h or 24 h post-transfection. When we increased the plasmid amount to 50 ng-100ng, green fluorescence could be detected in a part of cells 12 hours post-transfection, and nucleolus localization could be observed in almost every GFP positive cell. These findings indicate that Tat being localized to the nucleolus is unlikely to be determined by its expression level.

Figures for referee with unpublished data and its description has been removed upon request by the authors.

Page 3 please check the following sentence:

Following HIV-1 infection of cells, LARP7 puncta, LARP7 staining signals displayed a markedly higher fluorescence intensity than the surrounding areas, were significantly induced both in number and their fluorescence intensity.

Response:

Thanks for the advice, we rewrote the sentence and revised the manuscript. The updated text is as follows:

Following HIV-1 infection, LARP7 puncta exhibited a significant increase in both number and fluorescence intensity. These droplets persisted for approximately 24 h post-infection and then began to transit to a more homogeneous morphology (Figures 1A and 1B).

Page 4-5: Sentence needs to be corrected: We found that under certain circumstance, the LARP7/Tat ratio higher, the lower and slower the HIV-1 transcription will be.

Response:

Thanks for the suggestion, we rewrote the sentence and revised the manuscript. The updated text is as follows:

We observed that under certain conditions, a higher LARP7/Tat ratio correlates with reduced HIV-1 transcription.

Page 5: Sentence should be corrected: These results suggest that LARP7/Tat ratio regulate their co-phase separation, we thus wondered whether it can also regulate their function on HIV-1transcription.

Response:

Thanks for the suggestion, we rewrote the sentence and revised the manuscript. The updated text is as follows:

These findings suggest that a higher LARP7/Tat ratio enhances their co-liquid liquid phase separation, prompting further investigation into its impact on HIV-1 transcription.

Referee #2:

Li et al have revised the manuscript and the new data adds important details. In particular, the RIP experiments in Figure 4E and the co-IP experiments in Figure 4F add important details that the lysine mutations not only affect condensate formation but also affect binding to the components of the complex. The revisions have satisfied my major comments.

Response:

We really appreciate the referees' comments and constructive suggestions on our work.

Weihua Li
State Key Laboratory of Proteomics, Institute of Basic Medical Sciences, China National Center of Biomedical Analysis
Beijing 100850
China

Dear Dr. Li,

I am very pleased to accept your manuscript for publication in the next available issue of EMBO reports. Thank you for your contribution to our journal.

Yours sincerely,
